# Small-molecule-biased formyl peptide receptor agonist compound 17b protects against myocardial ischaemia-reperfusion injury in mice

Cheng Xue Qin[1,2,*], Lauren T. May[3,*], Renming Li[1,2], Nga Cao[1], Sarah Rosli[1], Minh Deo[1], Amy E. Alexander[1], Duncan Horlock[1], Jane E. Bourke[4], Yuan H. Yang[5], Alastair G. Stewart[2], David M. Kaye[1], Xiao-Jun Du[1], Patrick M. Sexton[3], Arthur Christopoulos[3], Xiao-Ming Gao[1,**] & Rebecca H. Ritchie[1,2,6,**]

Effective treatment for managing myocardial infarction (MI) remains an urgent, unmet clinical need. Formyl peptide receptors (FPR) regulate inflammation, a major contributing mechanism to cardiac injury following MI. Here we demonstrate that FPR1/FPR2-biased agonism may represent a novel therapeutic strategy for the treatment of MI. The small-molecule FPR1/FPR2 agonist, Compound 17b (Cmpd17b), exhibits a distinct signalling fingerprint to the conventional FPR1/FPR2 agonist, Compound-43 (Cmpd43). In Chinese hamster ovary (CHO) cells stably transfected with human FPR1 or FPR2, Compd17b is biased away from potentially detrimental FPR1/2-mediated calcium mobilization, but retains the pro-survival signalling, ERK1/2 and Akt phosphorylation, relative to Compd43. The pathological importance of the biased agonism of Cmpd17b is demonstrable as superior cardioprotection in both *in vitro* (cardiomyocytes and cardiofibroblasts) and MI injury in mice *in vivo*. These findings reveal new insights for development of small molecule FPR agonists with an improved cardioprotective profile for treating MI.

[1] Baker IDI Heart and Diabetes Institute, 75 Commercial Road, Melbourne, Victoria 3004, Australia. [2] Department of Pharmacology, University of Melbourne, Melbourne, Victoria 3010, Australia. [3] Department of Pharmacology, Division of Drug Discovery Biology, Department of Pharmacology, Monash University, 381 Royal Parade, Parkville, Victoria 3052, Australia. [4] Department of Pharmacology, Monash University, Wellington Road and Blackburn Road, Clayton, Victoria 3800, Australia. [5] Centre of Inflammatory Diseases, Monash University, Clayton, Victoria 3168, Australia. [6] Department of Medicine, Monash University, Parkville, Victoria 3800, Australia. * These authors contributed equally to this work. ** These authors jointly supervised this work. Correspondence and requests for materials should be addressed to R.H.R. (email rebecca.ritchie@bakeridi.edu.au).

There remains a current paucity of effective pharmacological treatments for myocardial infarction (MI) and post-MI cardiac remodelling, beyond early interventional coronary revascularization and thrombolytic procedures. As a result, MI and subsequent heart failure (HF) are the leading global causes of death, accounting for more than 7 million deaths annually[1,2]. The mechanisms of MI injury are multifactorial, with infiltration of circulating leucocytes and cardiomyocyte $Ca^{2+}$ overload representing the key causal contributors to loss of myocardial viability[3]. Development of innovative pharmacotherapies targeting MI injury is thus essential to address this unmet clinical need[4].

The FPR family of G protein-coupled receptors (GPCRs) plays an important role in host defence, regulation of inflammation and its resolution[3,5,6]. FPRs thus represent a novel therapeutic target in MI, where inflammation is a major contributing mechanism[3,5–8]. Endogenous non-selective FPR agonists annexin-A1 and its N-terminal-peptide Ac2-26 confer significant cardioprotection early after myocardial I–R injury[7,9–12]. Three FPR subtypes are expressed in humans; FPR1 and FPR2 (previously known as aspirin-triggered lipoxin receptor/FPR-like receptor-1 (FPRL1))[13] are widely distributed. FPR2 has been considered largely responsible for attenuation of inflammation[6,13], whereas cardiomyocyte–FPR1 promotes cardiomyocyte survival and preservation of left ventricular (LV) function[8]. Dual FPR1/FPR2 agonists may thus offer significant therapeutic potential for attenuating MI injury.

An important distinguishing feature of FPRs is their ability to interact with multiple, structurally diverse ligands (lipids, proteins and peptides) that can stimulate opposing cellular responses downstream of receptor activation[3], which appear specific to both the cell type and the ligand[6]. For example, annexin-A1, lipoxin $A_4$ (LxA$_4$) and Ac2-26 exhibit anti-inflammatory and pro-resolving actions[6,14,15]. In direct contrast, other FPR agonists bacterial-derived N-formyl-Met-Leu-Phe (fMLP), serum amyloid-A, Trp-Lys-Tyr-Met-Val-D-Met-NH$_2$ and Leu-Glu-Ser-Ile-Phe-Arg-Ser-Leu-Leu-Phe-Arg-Val-Met promote pro-inflammatory signal transduction[16]. These findings cannot be reconciled with differences in FPR subtype selectivity and instead are suggestive of biased agonism. Biased agonism reflects the ability of GPCRs to adopt different conformational states, each linked to distinct cellular outcomes[17–20]. Biased agonism hence provides the opportunity to potentially promote desired on-target signal transduction free of on-target adverse effects. This emerging paradigm is currently revolutionizing drug discovery, including in the context of acute HF[21], but is yet to be exploited post-MI in vivo. Development of small-molecule FPR agonists (potentially more resistant to degradation than peptides) have attracted significant attention as potential therapeutics for a range of inflammatory disorders[3]; however, this is yet to be extended to their prospective cardioprotective effects.

The objectives of this study were to (i) determine whether small-molecule FPR agonists exhibit different signalling fingerprints at FPR1 and FPR2 in vitro, comparing the small-molecule Amgen compound-43 (Cmpd43)[22], and N-(4-Bromophenyl)-2-[5-(3-methoxybenzyl)-3-methyl-6-oxo-6H-pyridazin-1-yl]-propionamide identified as compound 17b (Cmpd17b, Supplementary Fig. 1)[23]; (ii) determine the effect of Cmpd43 and Cmpd17b on cardiac injury responses in isolated cardiomyocytes and cardiofibroblasts; (iii) determine the impact of both on multiple myocardial ischaemic injury responses across four different time points in mice in vivo.

Our results reveal that, relative to Cmpd43, Cmpd17b signalling is markedly (30-fold) biased away from intracellular $Ca^{2+}$ mobilization ($Ca_i^{2+}$) yet stimulation of cell survival reperfusion injury salvage kinase pathways (RISK) such as extracellular signal-regulated kinase 1/2 (ERK1/2) and Akt phosphorylation remains intact. Both agents are revealed as dual FPR1/FPR2 agonists, in contrast to their initial description as FPR1- (Cmpd17b)[23] and FPR2-selective (Cmpd43)[22], respectively. Notably, the contrasting bias profiles for the two FPR agonists are associated with differential pathophysiological benefit. Cmpd17b mediates a range of cardioprotective effects up to 7 days post ischaemia-reperfusion (I–R) in vivo, with similar cardioprotection in vitro. In contrast, Cmpd43 is devoid of such cardioprotection. To our knowledge, these findings represent the first quantification of FPR-biased agonism, and reveal compelling new insights for development of small-molecule FPR agonists for treating MI.

## Results

**FPR agonists Cmpd17 and Cmpd43 activate hFPR1 & hFPR2.** The functional properties of FPR agonists, fMLP, Cmpd17b and Cmpd43, were assessed in Flp-In-Chinese hamster ovary (CHO) cells stably expressing human FPR1 (hFPR1-CHO), human FPR2 (hFPR2-CHO) or human FPR3 (hFPR3-CHO). Detection of biased signalling between FPR agonists requires quantification at multiple intracellular signalling intermediates.

The time course of ERK1/2 phosphorylation was assessed for each FPR agonist at each FPR subtype (Supplementary Fig. 2). During these assays, hFPR-CHO cells were exposed to fMLP, Cmpd17b and Cmpd43 (each 10 μM) in serum-free DMEM at 37 °C in 5% $CO_2$. Maximal ERK1/2 phosphorylation was observed after 5 min exposure to fMLP and Cmpd43 in both hFPR1 and hFPR2-CHO cells, whereas 7 min was the time point of maximal effect of Cpmd17b. None of the three FPR agonists elicited effects in hFPR3-CHO cells, demonstrating their lack of functional responses at FPR3, and consistent with absence of native Fpr1 or Fpr2 expression in CHO cells. An alternative GPCR agonist, adenosine, did not elicit a response in hFPR1-CHO cells, providing further support that the responses observed in hFPR1-CHO cells to fMLP, Cmpd17b and Cmpd43 are specific to FPR agonists and are mediated through FPR1.

Our results show that both Cmpd17b and Cmpd43 stimulated a concentration-dependent phosphorylation of key cardiomyocyte survival pathways ERK1/2, Akt1/2/3(Thr308) and Akt1/2/3(Ser474), and an increase in $Ca_i^{2+}$ (key to both $Ca^{2+}$-overload-induced cardiomyocyte loss post I–R as well as inflammatory cell migration) in hFPR1-CHO (Fig. 1) and hFPR2-CHO (Fig. 2) cells. Both Cmpd17b and Cmpd43 inhibited forskolin-stimulated cAMP accumulation in hFPR1-CHO and hFPR2-CHO cells, consistent with FPR1/2 coupling to $G_{i/o}$ proteins and inhibiting adenylate cyclase. However, a bi-phasic concentration-response curve was evident for the influence of Cmpd43 on forskolin-stimulated cAMP accumulation, switching from an inhibitory to a stimulatory response at higher concentrations. Under these circumstances, potency (pEC$_{50}$) values were derived over the lower concentration range, reflecting agonist-mediated inhibition of cAMP accumulation. At both FPR1 and FPR2, Cmpd43 had a higher potency, ~10-fold or greater, than Cmpd17b for each signalling pathway assessed (Supplementary Table 1).

For comparison, the high-affinity FPR1-selective peptide agonist fMLP stimulated concentration-dependent increases in $Ca_i^{2+}$, inhibition of cAMP accumulation and phosphorylation of ERK1/2, Akt1/2/3(Thr308) and Akt1/2/3(Ser474) in hFPR1-CHO cells, with higher (nanomolar) potency but similar maxima to Cmpd17b and Cmpd43 (Fig. 1 and Supplementary Table 1). Similar to Cmpd43, fMLP also exhibited a bi-phasic concentration-response curve on forskolin-stimulated cAMP accumulation, switching from an inhibitory to a stimulatory response at higher

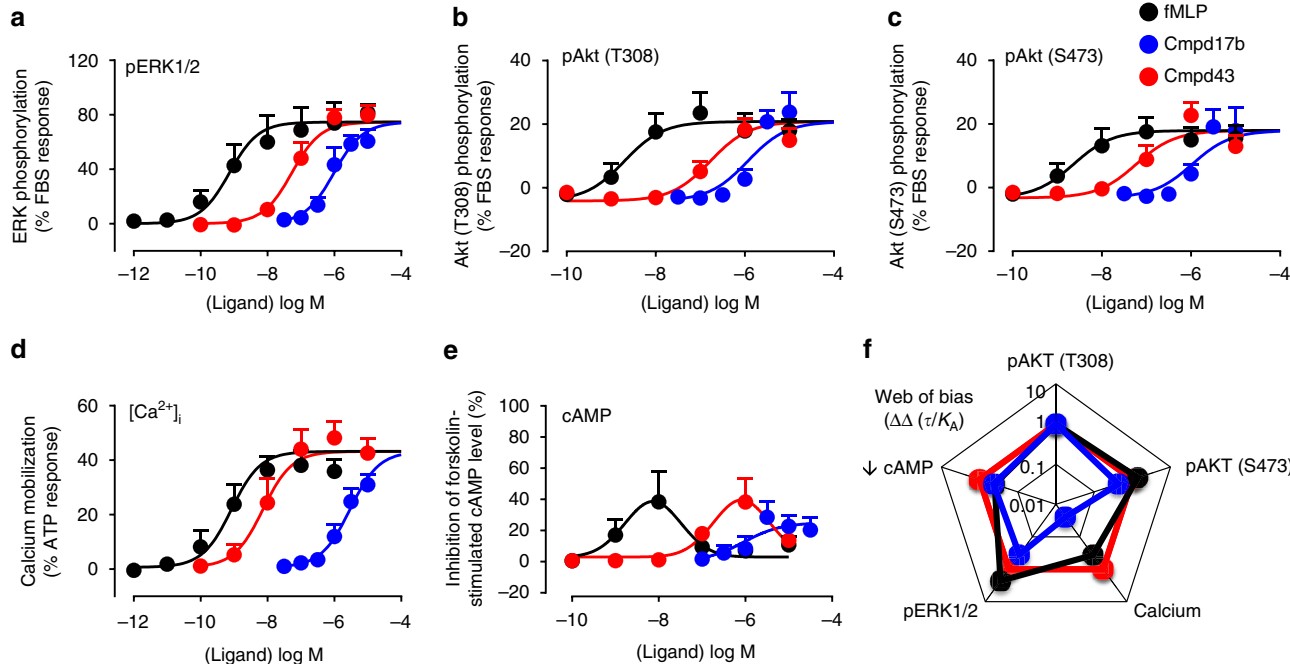

**Figure 1 | Signaling fingerprint of FPR agonists in CHO cells stably expressing hFPR1 reveals that Cmpd17b is a biased agonist at FPR1.** Influence of fMLP (black), Cmpd17b (blue) and Cmpd43 (red) on intracellular signalling intermediates downstream of GPCRs. (**a**) ERK1/2 phosphorylation ($n = 6$ separate experiments, each performed in triplicate); (**b**) Akt1/2/3(T308) phosphorylation ($n = 5$ separate experiments, each performed in triplicate); (**c**) Akt1/2/3(S473) phosphorylation ($n = 5$ separate experiments for Cmpd17b and Cmpd43, and $n = 4$ separate experiments for fMLP, each performed in triplicate); (**d**) $Ca_i^{2+}$ ($n = 4$ separate experiments for Cmpd17b and Cmpd43, and $n = 3$ separate experiments for fMLP, each performed in triplicate); and (**e**) inhibition of forskolin-stimulated cAMP accumulation ($n = 4$ separate experiments for Cmpd17b, and $n = 3$ separate experiments for Cmpd43 and fMLP, each performed in triplicate). Results are eppressed as mean ± s.e.m. (**f**) The 'web of bias' showing bias factors (DDt/KA) normalized to Cmpd43 and reference pathway pSKT/2/3(T308), derived from results in **a–e**, expressed as the mean.

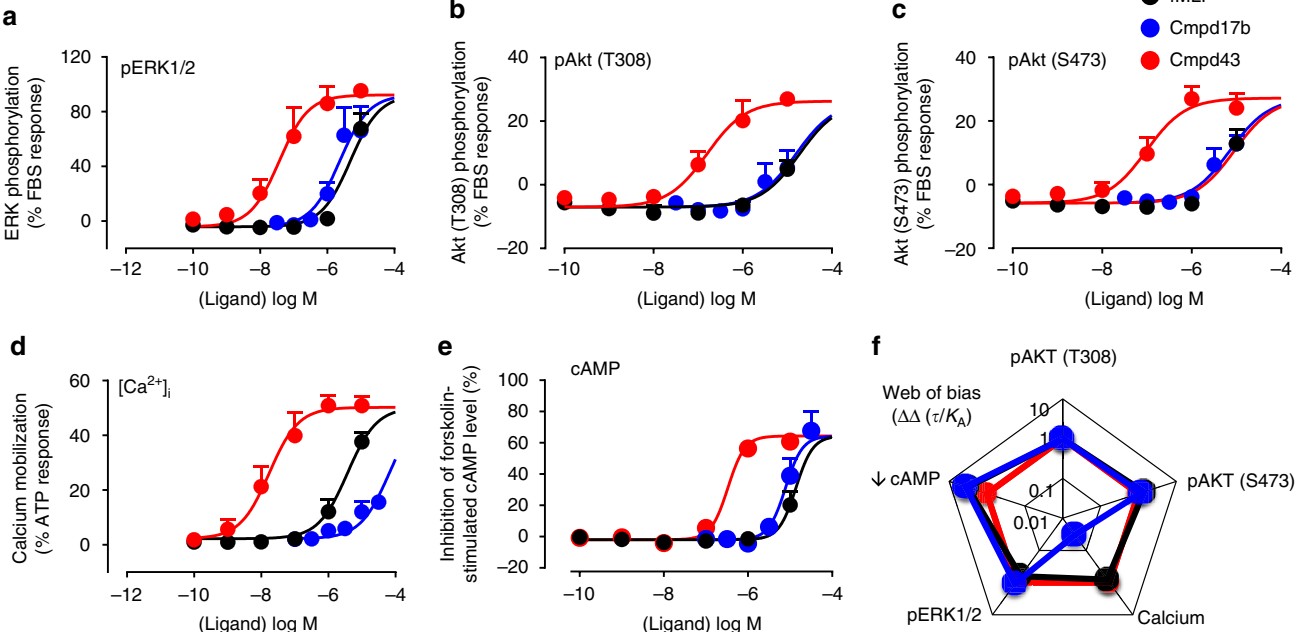

**Figure 2 | Signaling fingerprint of FPR agonists in CHO cells stably expressing hFPR2 reveals that Cmpd17b is a biased agonist at FPR2.** Impact of fMLP (black), Cmpd17b (blue) and Cmpd43 (red) on intracellular signalling intermediates downstream of GPCRs. (**a**) ERK1/2 phosphorylation ($n = 4$ separate experiments, each performed in triplicate); (**b**) Akt1/2/3(T308) phosphorylation ($n = 5$ separate experiments, each performed in triplicate); (**c**) Akt1/2/3(S473) phosphorylation ($n = 5$ separate experiments for Cmpd17b, and $n = 4$ separate experiments for Cmpd43 and fMLP, each performed in triplicate); (**d**) $Ca_i^{2+}$ ($n = 5$, 4 and 3 separate experiments for Cmpd17b, Cmpd43 and fMLP, respectively, each performed in triplicate); and (**e**) inhibition of forskolin-stimulated cAMP accumulation ($n = 4$ separate experiments for Cmpd17b and Cmpd43, and $n = 3$ separate experiments for fMLP, each performed in triplicate). Results are expressed as mean ± s.e.m. (**f**) The 'web of bias' showing bias factors ($\Delta\Delta\tau/K_A$) normalized to Cmpd43 and reference pathway pAkt1/2/3(T308), derived from results in **a–e**, expressed as the mean.

concentrations. None of the agonists tested elicited ERK phosphorylation in hFPR3-CHO cells (Supplementary Fig. 2), confirming specificity of the observed responses to the relevant hFPR1/hFPR2 subtype.

**Cmpd17b is a biased FPR agonist.** GPCRs can isomerize between a spectrum of conformational states, each signalling response reflecting a particular subset of these. To assess potential for ligand bias at a given GPCR, it is imperative that multiple signalling responses be incorporated into quantitative analysis; bias cannot be established on the basis of a single response[24]. In the present study, an operational model of agonism was fitted to concentration-response curves for each of the five signalling pathways. This generated a transduction coefficient ($\tau/K_A$) for each agonist at each signalling end point, a composite value of ligand efficacy ($\tau$) and affinity ($K_A$) that provides a measure of the overall agonist strength for coupling to the transduction pathway[24]. The transduction coefficient normalized for the system-dependent coupling efficiency for each pathway, $\Delta\text{Log}(\tau/K_A)$, reveals that Cmpd17b is significantly biased away from $Ca_i^{2+}$ at both hFPR1 and hFPR2 ($^{\wedge\wedge}P < 0.01$, $Ca_i^{2+}$ relative to all other signalling pathways, one-way analysis of variance (ANOVA) followed by Tukey's *post hoc* test; Supplementary Table 1). Bias factors, $\Delta\Delta\tau/K_A$ values, can be visualized on the 'web of bias', which clearly demonstrates that relative to Cmpd43, Cmpd17b was ~30-fold biased away from $Ca_i^{2+}$ at both FPR1 (Fig. 1) and FPR2 (Fig. 2).

**FPR agonist Cmpd17b reduces cardiac cell injury responses.** The direct effects of Cmpd17b and Cmpd43 on cell injury

responses *in vitro* in primary cardiomyocytes and cardiofibroblasts were determined, which express native *Fpr* subtypes (Supplementary Fig. 3). We demonstrated that Cmpd17b rescued cardiac troponin I (cTnI) release post-simulated I–R *in vitro*, in both rat and mouse cardiomyocytes (Fig. 3). Furthermore, Cmpd17b and Cmpd43 exhibited directly contrasting effects on cardiomyocyte $Ca_i^{2+}$ responses, inhibition versus stimulation, respectively (Fig. 3). Cardiofibroblasts were subjected to 24 h pro-fibrotic transforming growth factor (TGF)-β stimulation. The pro-fibrotic connective tissue growth factor (*CTGF*) response to TGF-β was significantly attenuated in Cmpd17b-treated, but not Cmpd43-treated, cardiofibroblasts relative to control (Fig. 3). Similarly, TGF-β-induced pro-inflammatory interleukin (IL)-1β responses tend to be reduced in Cmpd17b-treated but not Cmpd43-treated cardiofibroblasts (Fig. 3, $P = 0.06$, one-way ANOVA with Dunnet's *post hoc* test).

The International Union of Basic and Clinical Pharmacology nomenclature for FPRs uses uppercase letters for human formyl peptide receptors (FPR, for example, FPR1), whereas title case is used for rodent receptors (for example, Fpr1)[13]. The relative gene expression of the FPR family in cardiomyocytes is *rFpr1* > *rFpr2* >> *rFpr3*; this is similar to cardiofibroblasts, except *mFpr3* is not detected (assessed via real-time PCR, Supplementary Fig. 3). The impact of treatment with FPR agonists Cmpd17b and Cmpd43 on cellular *Fpr1/2* gene expression was also determined in cardiomyocytes and cardiofibroblasts. Agonist treatment elicited minimal impact on *Fpr* gene expression in either cardiac cell type, with the exception of the trend for Cmpd43 to reduce cardiomyocyte *rFpr2* ($P = 0.05$, one-way ANOVA with Dunnet's *post hoc* test) and to blunt the TGF-β-mediated reduction in cardiofibroblast *mFpr1* (Supplementary Fig. 3).

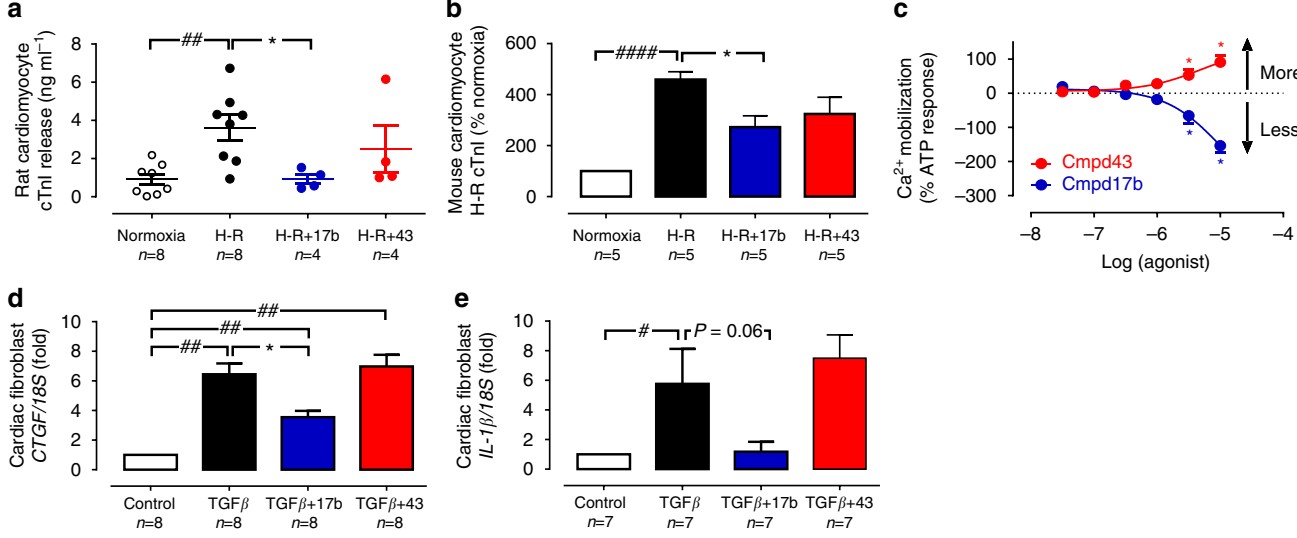

**Figure 3 | Protective actions of FPR agonist Cmpd17b against cardiomyocyte injury responses *in vitro*. (a)** Effect of small-molecule FPR agonists (both 1 μM, added on reoxygenation) on rat cardiomyocyte cTnI release subsequent to H–R; $^{\#\#}P < 0.01$ versus sham and $^*P < 0.05$ versus vehicle-treated H–R in paired cardiomyocytes from the same preparation (one-way ANOVA, Dunnett's *post hoc*, $n = 4$ separate cardiomyocyte preparations). **(b)** Effect of small-molecule FPR agonists (both 1 μM, added on simulated reperfusion) on mouse cardiomyocyte cTnI release subsequent to simulated I–R; $^{\#\#\#\#}P < 0.0001$ versus sham and $^*P < 0.05$ versus vehicle-treated H–R in paired cardiomyocytes from the same preparation (one-way ANOVA, Dunnett's *post hoc*, $n = 5$ separate cardiomyocyte preparations). **(c)** Concentration-dependent $Ca_i^{2+}$ responses to small-molecule FPR agonists Cmpd 17b ($n = 4$) and Cmpd43 ($n = 5$) in primary rat cardiomyocytes; $^*P < 0.05$ versus lowest concentration studied (two-way repeated measures ANOVA, Tukey's *post hoc*). Effect of Cmpd17b and Cmpd43 (both 10 μM) on TGF-β stimulation of cardiofibroblast. **(d)** Pro-fibrotic *CTGF* and **(e)** pro-inflammatory *IL-1β* expression; $^{\#}P < 0.05$, $^{\#\#}P < 0.01$ versus sham and $^*P < 0.05$ versus TGF-β stimulation in paired cardiofibroblasts from the same preparation, on one-way ANOVA with Dunnett's *post hoc* test ($n$ per group indicated below the x axis). Results expressed as mean ± s.e.m. Controls (open symbols/bars); H–R or TGF-β (black symbols/bars); Cmpd17b (blue symbols/bars); Cmpd43 (red symbols/bars).

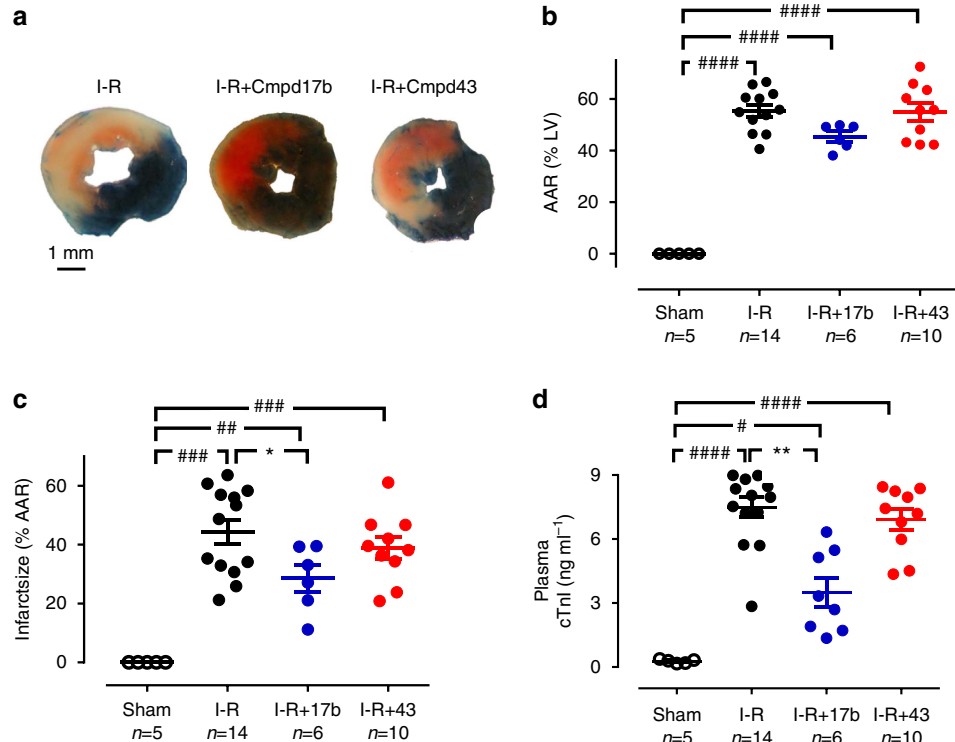

**Figure 4 | FPR agonist Cmpd17b reduces cardiac necrosis 24 h post I–R injury in vivo.** (**a**) Representative 2,3,5-triphenyltetra-zolium chloride (TTC)-stained LV transverse slices after 24 h reperfusion in mice allocated to vehicle, Cmpd17b or Cmpd43 (both 50 mg kg$^{-1}$ i.p.)-treatment groups (scale bar, 1 mm). Areas stained dark blue, white and red represented non-risk, infarcted and ischaemic but non-infarcted zones, respectively. Pooled data for (**b**) AAR (calculated as total infarcted plus ischaemic but non-infarcted zones, % total LV). (**c**) Myocardial infarct size (%AAR) and (**d**) plasma cTnI levels. Results are expressed as mean ± s.e.m., with *n* (number of mice) per group indicated below the *x* axis. $^{\#}P < 0.05$, $^{\#\#}P < 0.01$, $^{\#\#\#}P < 0.001$ and $^{\#\#\#\#}P < 0.0001$ versus sham; $^{*}P < 0.05$ and $^{**}P < 0.01$ versus vehicle-treated I–R on one-way ANOVA with Dunnett's *post hoc* test. Shams (open symbols); I–R (black symbols); Cmpd17b (blue symbols); Cmpd43 (red symbols).

**FPR agonist Cmpd17b reduces early cardiac necrosis in vivo.** To assess the impact of the small-molecule FPR agonists in an animal model of myocardial I–R *in vivo*, C57BL/6 mice underwent reversible ligation of left arterial descending (LAD) coronary artery followed by reperfusion[25]. Mice were randomly assigned to three I–R cohorts (24, 48 h and 7-day reperfusion groups) to measure different functional outcomes. First, the impact of FPR agonists (50 mg kg$^{-1}$, intraperitoneal (i.p.), administered just before 24 h reperfusion) on cardiac necrosis after 24 h were assessed. There were no differences between I–R groups in the LV identified as the area-at-risk (AAR, ∼60% on Evans blue) after 24 h (Fig. 4). Of the AAR in vehicle-treated I–R mice, ∼40% was infarcted (Fig. 4 and Supplementary Fig. 4) and plasma levels of cTnI were markedly elevated. We demonstrated that Cmpd17b significantly reduced both infarct size (IS, by ∼35%) and cTnI levels (Fig. 4). In contrast, Cmpd43 lacked cardioprotective actions compared with vehicle-treated I–R mice.

**Cmpd17b attenuates cardiac and systemic responses after 48 h.** We assessed the extent of I–R-induced cardiac injury and systemic inflammation after 48 h reperfusion *in vivo*, and the impact of FPR agonists, via immunofluorescent detection of LV neutrophil (anti-Ly-6B.2) and macrophage (anti-CD68) content. Cmpd17b significantly attenuated I–R-induced increases in LV neutrophil density compared with vehicle-treated I–R mice, by almost 50% (Fig. 5 and Supplementary Fig. 5); this cardioprotective action was absent in Cmpd43-treated I–R mice. In contrast, although LV macrophage density was also significantly

elevated in vehicle-treated I–R mice compared with sham, this was not significantly blunted by either FPR agonist (Supplementary Fig. 6). The end point plasma concentrations for each FPR agonist were comparable after 24 versus 48 h reperfusion in mice *in vivo*, with Cmpd17b and Cmpd43 levels ∼20 h post-final dosing ∼1 and 0.2 μM, respectively. Further, the extent of LV cell death 48 h post I–R was blunted by Cmpd17b (but not Cmpd43; Fig. 5).

Circulating total white blood cells (WBCs) were significantly increased in vehicle-treated mice compared with sham (Fig. 6), with similar trends for circulating lymphocytes and neutrophils. A similar (but nonsignificant) tendency for increases in circulating levels of both pro-inflammatory IL-1β and monocytes was also evident, consistent with systemic inflammation. Cmpd17b significantly blunted I–R-induced increases in total WBC, circulating lymphocytes, neutrophil levels and IL-1β, without affecting circulating monocytes. In contrast, Cmpd43 significantly increased circulating monocytes and lymphocytes after 48 h reperfusion compared with sham. Together, our observations suggest that, in contrast to Cmpd43, Cmpd17b attenuates the early inflammatory response associated with reperfusion injury.

**Cmpd17b limits cardiac remodelling and dysfunction in vivo.** Next, we determined the impact of FPR agonists on I–R-induced cardiac cell death (by CardioTACS) and fibrosis (by picrosirius red) after 7-day reperfusion. Myocardial I–R significantly increased cardiac cell death and collagen deposition (Fig. 7 and Supplementary Fig. 7). Cmpd17b-treated I–R mice exhibited

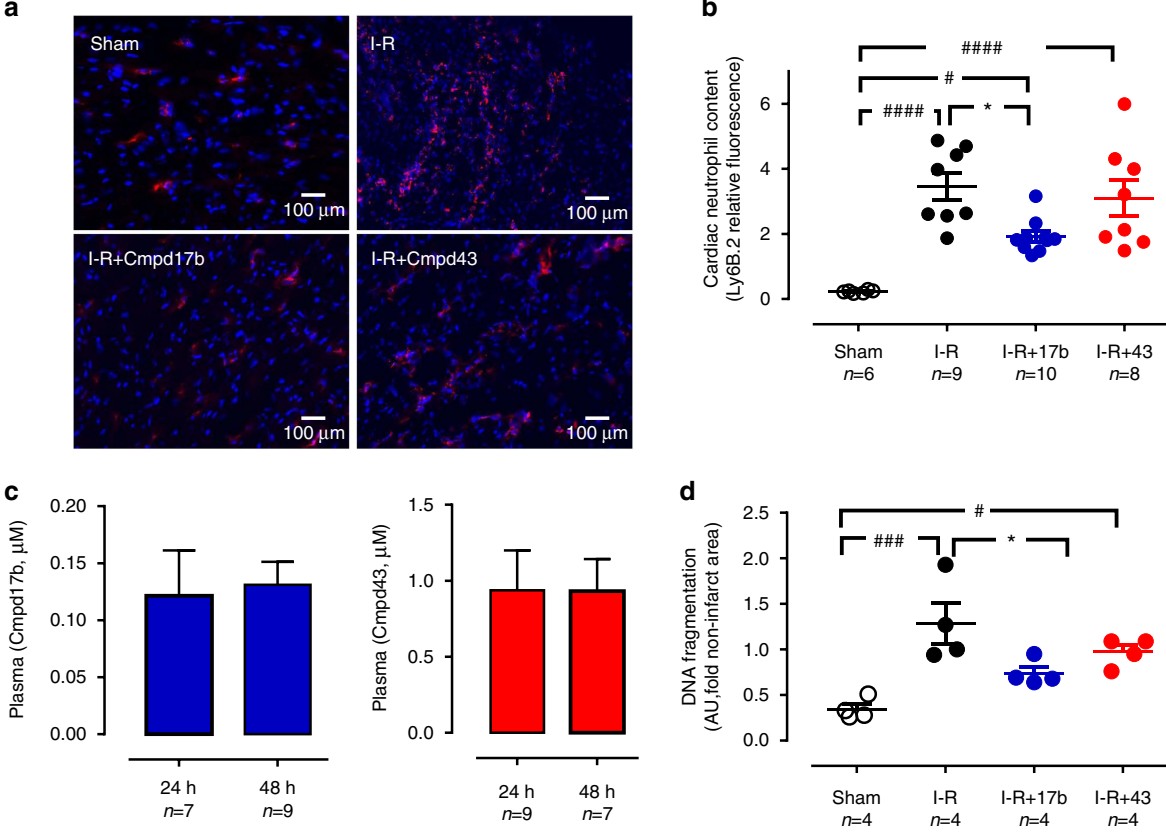

**Figure 5 | FPR agonist Cmpd17b reduces cardiac inflammation and injury 48 h post I–R.** (**a**) Representative immunofluorescent images of LV neutrophil content (using anti-Ly-6B.2 antibody) from sham, vehicle- and FPR agonist-treated (Cmpd17b or Cmpd43, both 50 mg kg$^{-1}$ per day, i.p.) mice, 48 h post I–R ($\times$ 40 magnification, scale bars, 100 μm). (**b**) Pooled data for LV Ly-6B.2-positive immunofluorescence. (**c**) End point plasma Cmpd17b and Cmpd43 concentrations after 24 and 48 h reperfusion in mice *in vivo*. (**d**) Quantification of dead:viable cells from CardioTAC-stained LV sections 48 h post I–R (expressed as fold sham). Results are expressed as mean ± s.e.m., with n (number of mice) per group indicated below the x axis. #$P < 0.05$, ###$P < 0.001$ and ####$P < 0.0001$ versus sham and *$P < 0.05$ versus vehicle-treated mice on one-way ANOVA with Dunnett's *post hoc* test. Shams (open symbols/bars); I–R (black symbols/bars); Cmpd17b (blue symbols/bars); Cmpd43 (red symbols/bars).

significantly attenuated LV cell death in the infarct area after 7-day reperfusion, comparable to levels in sham ($P = $ NS (non-significant) versus sham, Fig. 7, one-way ANOVA with Dunnet's *post hoc* test). The impact of I–R on LV fibrosis in this region persisted in Cmpd43-treated I–R mice; however, cardiac collagen deposition in Cmpd17b-treated mice was not significantly different from either sham or vehicle I–R (Fig. 7). Further, Cmpd17b significantly reduced heart and LV weights in I–R mice, comparable to sham (normalized to bodyweight (BW), both $P = $ NS versus sham Fig. 7 and Supplementary Table 2, one-way ANOVA with Dunnet's *post hoc* test). Cmpd43 again lacked these cardioprotective actions compared with vehicle-treated I–R mice.

The impact of Cmpd17b on ischaemia-induced cardiac dysfunction and inflammatory gene expression was assessed 4 weeks after permanent LAD occlusion (that is, in the setting of MI rather than I–R). Fractional shortening (FS), LV ejection fraction and stroke volume were all significantly impaired 7-day post MI, and remained impaired after 4 weeks post MI (Fig. 8). Cmpd17b significantly improved FS by 4 weeks post MI, with similar trends on both LV ejection fraction and stroke volume (both $P = 0.06$ Cmpd 17b-treated versus vehicle-treated MI; two-way ANOVA with Tukey's *post hoc* test; Fig. 8). Furthermore, Cmpd 17b significantly ameliorated the MI-induced increase in LV gene expression of *Fpr1, Fpr2, CD68* and *TNF-α* in the infarcted zone of the LV (Fig. 8).

**Systemic and cardiac characteristics following I–R injury in vivo.** Body and organ weights measured after 48 h and 7-day reperfusion are shown in Supplementary Table 2. Following 48 h post-ischaemic reperfusion, no differences in body, atria or lung weights were observed. Wet heart weight (HW) was, however, significantly increased in vehicle- and Cmpd17b-treated I–R mice when normalized to BW. A similar but nonsignificant trend for HW:BW ratio was evident after 7-day reperfusion with vehicle-treated I–R compared with sham mice ($P = 0.07$, Supplementary Table 2). Interestingly, administration of Cmpd17b (50 mg kg$^{-1}$ per day i.p.) significantly reduced both total HW, and LV weight normalized to BW, consistent with protective actions of Cmpd17b on cardiac remodelling post I–R. This cardioprotection was, however, not shared by Cmpd43 at the same dosing regimen.

End point body and organ weights from mice allocated to cohort 4, measured after 4 weeks MI, are shown in Supplementary Table 3. Additional echocardiographic analysis of LV function in anaesthetized mice at study end point, complementing that provided in Fig. 8, shows that increases in heart, LV, atrial and lung weight normalized to BW were observed in addition to increased LV chamber dimensions left ventricular end systolic diameter (LVESD) and left ventricular end diastolic diameter (LVEDD) (Supplementary Table 3). Importantly, Cmpd17b significantly blunted the MI-induced lung weight:BW, consistent with reduced progression towards HF.

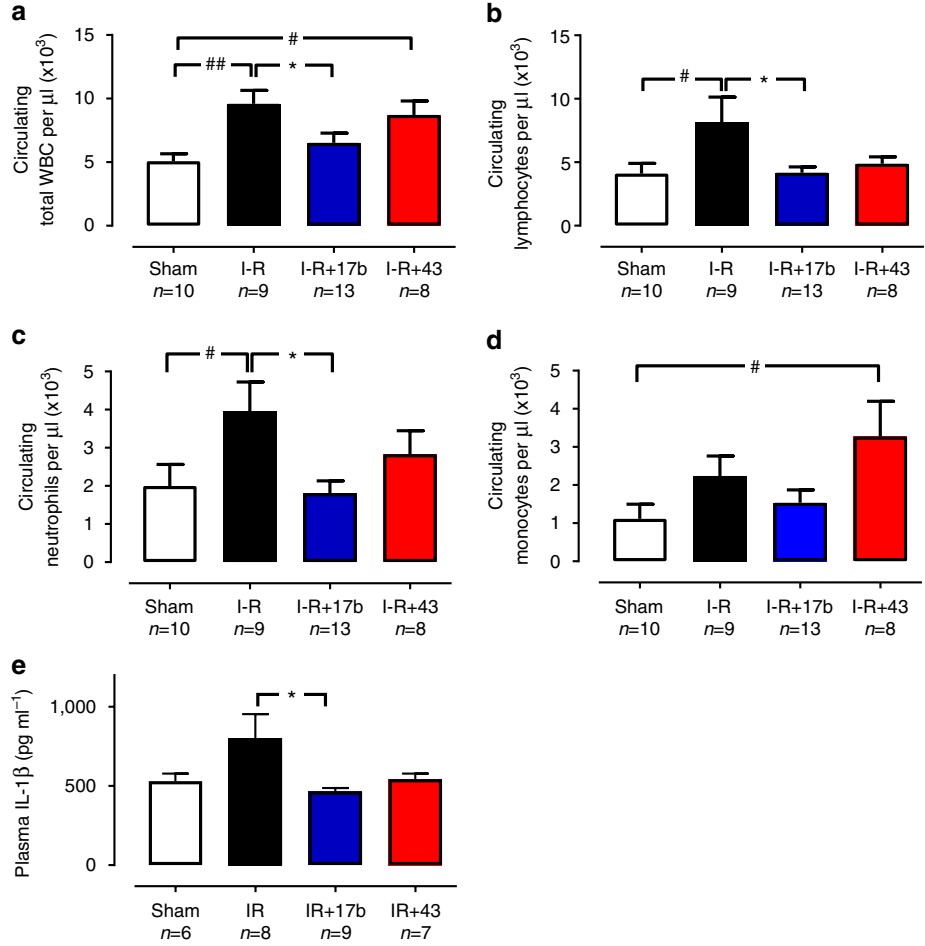

**Figure 6 | FPR agonist Cmpd17b reduces systemic inflammation 48 h-post I–R injury.** Circulating levels of (**a**) total WBCs, (**b**) lymphocytes, (**c**) neutrophils, (**d**) monocytes and (**e**) IL-1β in sham, vehicle- and FPR agonist-treated (Cmpd17b or Cmpd43, both 50 mg kg$^{-1}$ per day, i.p.) mice after 48 h reperfusion. Results are expressed as mean ± s.e.m. #$P < 0.05$, ##$P < 0.01$ versus sham,*$P < 0.05$ versus vehicle-treated mice on one-way ANOVA with Dunnett's *post hoc* test (*n*, number of mice per group, indicated below the *x* axis). Shams (open bars); I–R (black bars); Cmpd17b (blue bars); Cmpd43 (red bars).

## Discussion

Despite considerable drug discovery efforts to identify innovative pharmacotherapies for improving outcomes after MI, an urgent, unmet clinical need remains. We now demonstrate that FPR1/FPR2-biased agonism may represent one such novel strategy to limit MI. This breakthrough observation is the first to examine GPCR agonist bias in the context of myocardial cardioprotection *in vivo*.

Upon binding hFPR1 or hFPR2, Cmpd17b and Cmpd43 stimulated a range of biochemical events, including ERK1/2 phosphorylation, Akt1/2/3 phosphorylation, cAMP inhibition and increased $Ca_i^{2+}$. Our results in hFPR-CHO cells reveal that Cmpd17b and Cmpd43 activate both FPR1 and FPR2 at the level of ERK1/2-Akt signalling. In contrast to the initial reports suggesting FPR2 selectivity for Cmpd 43 (ref. 22) and FPR1 selectivity for Cmpd17b (ref. 23), our results in hFPR-CHO cells reveal that both agonists are clearly both active at FPR1 and FPR2 at the level of ERK1/2-Akt signalling. FPRs are inherently dynamic, interacting with multiple, structurally diverse, naturally occurring ligands that stimulate qualitatively different physiological responses with respect to inflammation and its resolution, dependent on both the cell type and the ligand[6,26]. This suggests that biased agonism may be an underappreciated biological mechanism for this receptor family.

Biased agonism posits that different agonists activating the same receptor can yield distinct signalling fingerprints in the same cell type, such that the resultant spectrum of effects is specific for individual ligands, rather than an agonist class effect[17,20]. To enable clustering of FPR agonists by their bias fingerprint, we quantified biased agonism at a number of judiciously chosen end points, particularly relevant to myocardial protection[19]. In cardiomyocytes, the RISK such as ERK1/2-Akt signalling are tightly linked to cell survival[27]. In contrast, increased $Ca_i^{2+}$ in the context of I–R injury is a contributing mechanism of cardiomyocyte damage (via $Ca^{2+}$ overload), and is a key contributor to influx of inflammatory neutrophils and macrophages into injured myocardium[3]. Statistical interrogation of biased agonism using the Black–Leff operational model revealed that Cmpd17b exhibited a significant, ~30-fold bias, away from $Ca_i^{2+}$, relative to Cmpd43, at both FPR1 and FPR2, while maintaining robust phosphorylation of ERK1/2 and Akt1/2/3. Importantly, a contrasting influence of Cmpd17b and Cmpd43 on $Ca^{2+}$ was also observed in primary cardiomyocytes, which express native *Fpr*.

To investigate the potential therapeutic efficacy of Cmpd17b over Cmpd43, this study utilized several myocardial I–R models to assess the cardioprotective effects of the two FPR agonists. Despite the higher relative potency of Cmpd43 over Cmpd17b across all the signalling pathways measured *in vitro*, the present

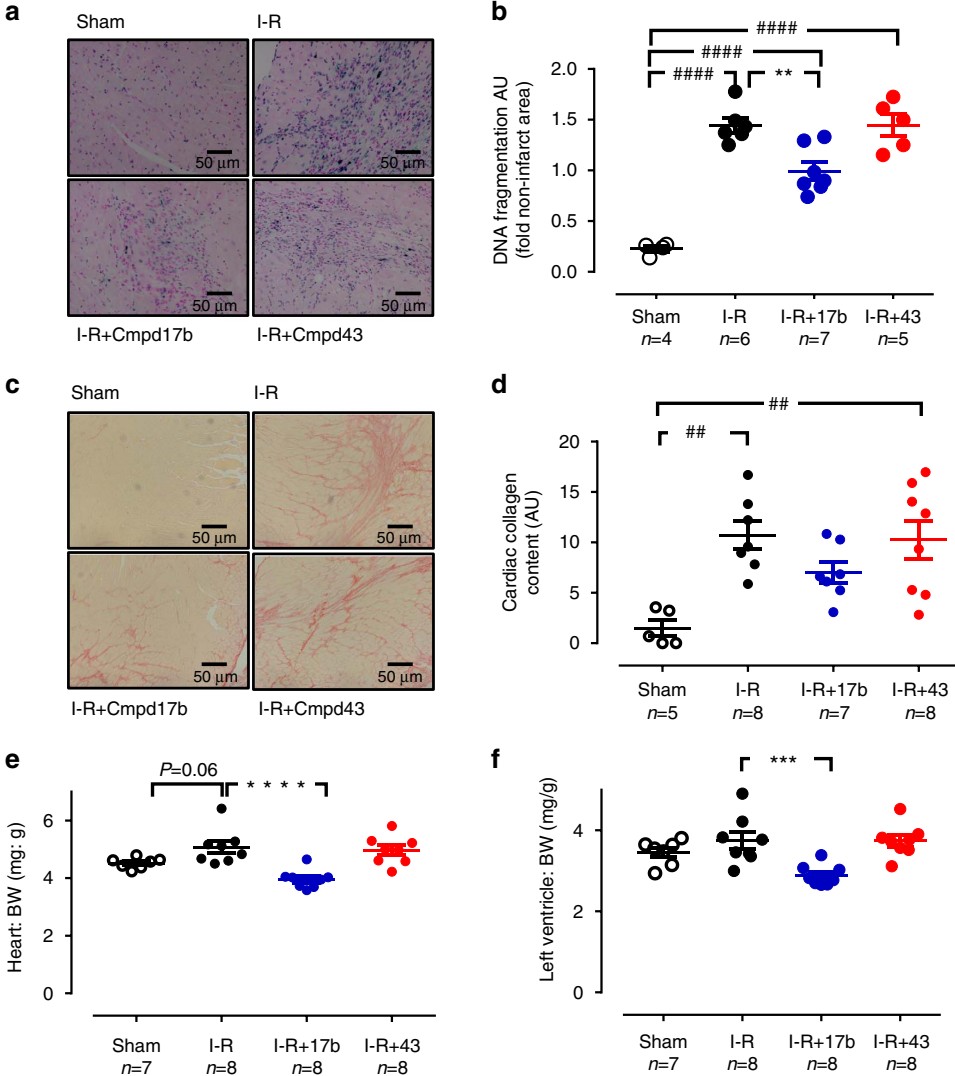

**Figure 7 | FPR agonist Cmpd17b reduces cardiac remodelling 7-day post I–R injury.** (**a**) Representative CardioTAC-stained LV sections 7-day post I–R; magnification × 200. (**b**) Quantification of dead:viable cells (expressed as fold sham). (**c**) Representative picrosirius red-stained LV cross-sections from sham, vehicle- and FPR agonist (Cmpd17b or Cmpd43, both 50 mg kg$^{-1}$ per day i.p.)-treated mice, 7-day post I–R. (**d**) Quantification of cardiac fibrosis (collagen appears red); magnification × 200. (**e**) Heart and (**f**) LV weight normalized to BW 7-day post I–R. Results are expressed as mean ± s.e.m. $^{##}P < 0.01$, $^{####}P < 0.0001$ versus sham; $^*P < 0.01$, $^{***}P < 0.001$, $^{****}P < 0.0001$ versus vehicle-treated mice on one-way ANOVA with Dunnett's *post hoc* test (*n*, number of mice per group, indicated below the *x* axis). Shams (open symbols); I–R (black symbols); Cmpd17b (blue symbols); Cmpd43 (red symbols).

study demonstrated that administration of Cmpd17b, but not Cmpd43, significantly reduce necrosis in isolated cardiomyocytes when subjected to hypoxia–reoxygenation (H–R), and attenuated gene expression of pro-fibrotic *CTGF* and pro-inflammatory *IL-1β* in isolated cardiofibroblasts when stimulated by TGF-β. Further, this beneficial effect was also translated to a preclinical model of myocardial I–R injury *in vivo*. Cmpd17b (administered just before reperfusion) significantly reduced early cardiac necrosis, consistent with promotion of cardiomyocyte survival, in contrast to Cmpd43 at the same dose. Accumulation of pro-inflammatory neutrophils into injured myocardium was again selectively reduced in Cmpd17b-treated mice, consistent with anti-inflammatory FPR-mediated responses following I–R[9–11]. Furthermore, our observations are consistent with Cmpd17b (but not Cmpd43), limiting I–R-induced cardiomyocyte apoptosis (and their replacement with cardiac collagen deposition, indicative of progression towards HF). Thus, the dual FPR1/FPR2-biased agonist Cmpd17b elicits superior

cardioprotection over Cmpd43. These findings cannot be reconciled with classical agonist efficacy and instead are suggestive of more complex mechanisms. On the basis of their opposing effects on primary cardiomyocyte $Ca_i^{2+}$, and their relative plasma levels *in vivo*, an insufficient dose or lower bioavailability of Cmpd43 relative to Cmpd17b does not provide an alternative explanation for its lack of cardioprotective actions. Furthermore, plasma levels of Cmpd43 were approximately fivefold those of Cmpd17b, yet Cmpd43 is a more potent agonist than Cmpd17b on the kinase survival pathways, as well as $Ca_i^{2+}$ *in vitro*. Collectively, our findings suggest that the differences in cardioprotection *in vivo* between Cmpd17b and Cmpd43 may be more readily explained by FPR-biased agonism at the level of $Ca_i^{2+}$. Relative to Cmpd43, Cmpd17b activation of cardioprotective signalling is not countered by concomitant potent increases in $Ca_i^{2+}$ (which in the case of Cmpd43 likely negate its potential confer cardioprotection). We hypothesized that if FPR agonists with biased signalling profiles towards (rather

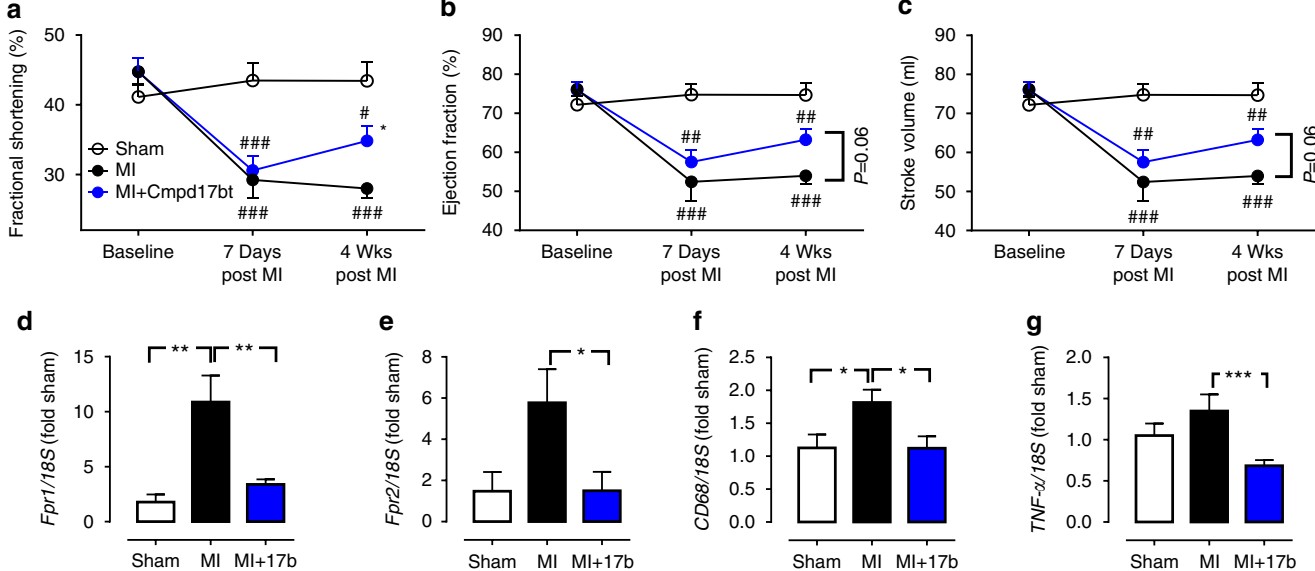

**Figure 8 | FPR agonist Cmpd17b reduces cardiac dysfunction and inflammation 4 weeks post MI.** (**a**) FS, (**b**) ejection fraction and (**c**) stroke volume at baseline, and 1 and 4 weeks post MI in sham ($n = 6$ mice, open symbols/bars), vehicle- ($n = 13$, black symbols/bars) and Cmpd17b ($n = 15$, blue symbols/ bars), 50 mg kg$^{-1}$ per day i.p.)-treated mice. #$P < 0.05$, ##$P < 0.01$, ###$P < 0.01$ versus sham; *$P < 0.05$ versus vehicle-treated MI mice on two-way ANOVA with Tukey's *post hoc* test. Impact on cardiac gene expression 4 weeks post MI in the infarcted LV MI, comparing sham ($n = 6$ mice), vehicle- ($n = 13$) and Cmpd17b ($n = 15$), including (**d**) *Fpr1*, (**e**) *Fpr2*, (**f**) *CD68* and (**g**) *TNF-α*; *$P < 0.05$, **$P < 0.01$, ***$P < 0.001$ on one-way ANOVA with Tukey's *post hoc* test. Results are expressed as mean ± s.e.m.

than away from) $Ca^{2+}$ are generated endogenously during MI, then Cmpd17b may act to block $Ca^{2+}$ overload by preventing receptor activation by such putative endogenous ligands. This is, however, beyond the scope of this study.

An ischaemic insult stimulates expansion and mobilization of haematopoietic stem progenitor cells to increase the supply of myeloid cells to the injured myocardium[28]. The resultant increased cardiac accumulation of neutrophils and macrophages likely contributes to myocardial damage following the insult[29], consistent with our observations 48 h after I–R. Cmpd17b blunted several components of this inflammatory response, including total circulating levels of WBC, neutrophils and IL-1β, in addition to cardiac neutrophil density. FPRs, particularly FPR2, are expressed on myeloid cells and on endothelium; their activation also regulates the inflammatory response and its resolution[3]. Within the current study, we have not determined the relative influence that Cmpd17b stimulation of FPRs expressed on the various cell types has on cardioprotection. We did not, for example, dissect out whether bias away from $Ca_i^{2+}$, downstream of FPRs, was also evident in myeloid cells.

Structure–function analyses of annexin-A1, serum amyloid-A and Cmpd43 have previously suggested that different FPR agonists may recognize topographically distinct binding domains, at FPR2 at least[30]. To date, such patterns of binding have not been considered for Cmpd17b, nor was binding at FPR1 evaluated. The nature of the different ligand–receptor interactions for structurally diverse FPR agonists are likely to dictate the downstream signal transduction observed, with different domains involved in regulation of $Ca^{2+}$ fluxes versus ERK1/2 phosphorylation. Mechanisms responsible for the differential signalling profile between Cmpd17b and Cmpd43 have not previously been elucidated. We speculate that Cmpd17b and Cmpd43 stabilize a distinct subset of FPR conformational states (illustrated schematically in Fig. 9), resulting in biased agonism. Clearly, Cmpd17b offers greater cardioprotection in the intact heart than Cmpd43, and the profile observed for Cmpd17b in both FPR-CHO cells and primary cardiomyocytes is consistent

with preferential coupling to cardioprotective signal transduction. However, given the potential for a system-dependent component of a bias profile, the exact mechanism of bias that leads to superior cardioprotection in models of MI warrants further investigation. Mitochondrial formylated-peptide ligands of FPR have been implicated in cardiovascular collapse in sepsis[31]; however, whether these ligands exhibit biased agonism is yet to be evaluated. Whether potential FPR dimerization contributes to cardioprotection *in vivo*, as has been suggested in transiently transfected FPR2-HEK293 cells *in vitro*[26], also remains to be elucidated.

In conclusion, the need for a balance between pro- and anti-inflammatory pathways to retain a homeostatic environment following an ischaemic insult has given rise to a great number of potential therapeutic options targeting the FPR family. Excitingly, our findings demonstrate for the first time that small-molecule FPR ligands exhibit biased signalling. Relative to the reference agonist Cmpd43, the biased agonist Cmpd17b favours markedly less $Ca_i^{2+}$ downstream of FPR1 and FPR2 relative to ERK1/2-Akt signal transduction; this bias away from detrimental $Ca_i^{2+}$ persisted in primary cardiomyocytes (Fig. 9). Further, we demonstrated that the biased FPR agonist Cmpd17b elicits superior outcomes following myocardial I–R injury in mice *in vivo*. This study has significantly improved understanding of the cardioprotective mechanisms of dual FPR agonists; selective activation of desirable signalling downstream of FPR activation may potentially avoid adverse consequences (Fig. 9). Our findings reveal potential new exciting strategies and routes of enquiry for targeted development of FPR-biased strategies for improving outcome after MI.

## Methods
**Assessment of biased signalling at human FPRs *in vitro*.** Small-molecule FPR agonists Cmpd17b and Cmpd43 (chemical structures and molecular weights depicted in Supplementary Fig. 1a) were synthesized by Anthem Bioscience (Bangalore, India) using a published synthetic route[22,23].

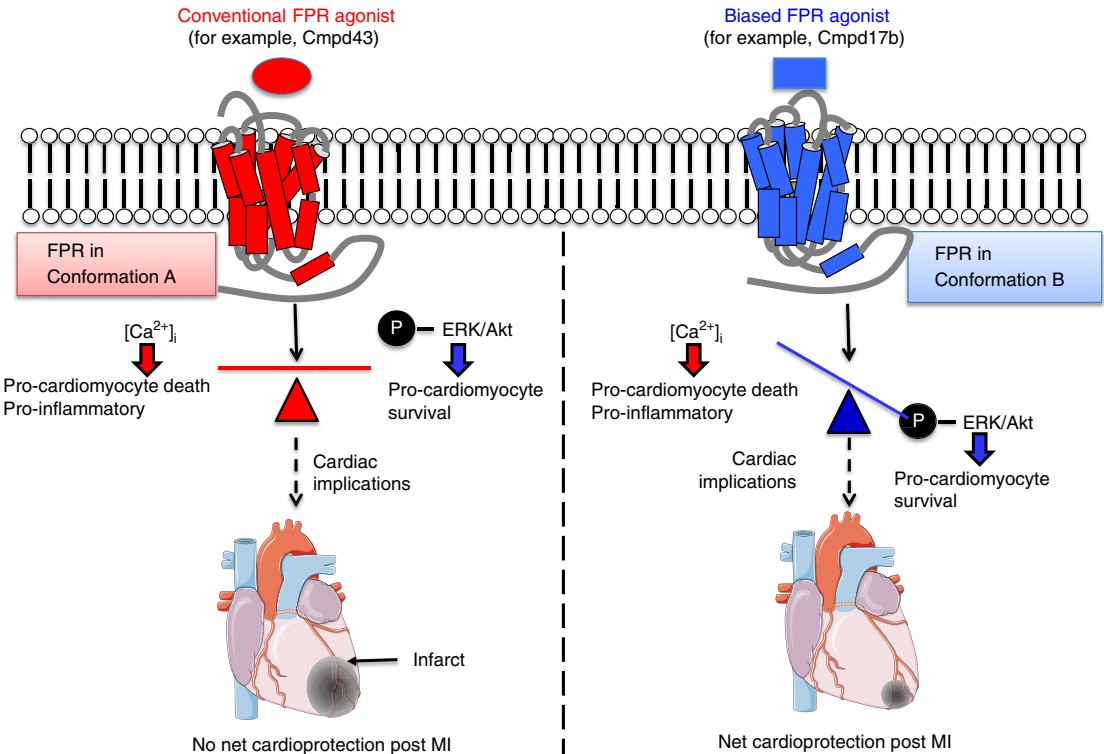

**Figure 9 | Summary of the bias properties of FPR agonists and implications for their role against myocardial I–R injury.** Cmpd17b and Cmpd43 activate both FPR1 and FPR2, but Cmpd17b offers superior cardioprotective effect, likely through biased activation of desirable cardioprotective RISK signalling pathways (ERK1/2, Akt1/2/3(T308), Akt1/2/3(S473)), while sparing pathways that may contribute undesirable consequences (for example, $Ca_i^{2+}$). Schema created using a modification to an image provided by Servier Medical Art by Servier (http://www.servier.com/Powerpoint-image-bank), licensed under a Creative Commons Attribution 3.0 Unported Licence (https://creativecommons.org/licences/by/3.0/).

*Generation and culture of stably transfected recombinant hFPR-CHO cells.* In the present study, FIp-IN CHO cells and Gateway plasmid were obtained from Invitrogen Inc. (Carlsbad, CA, USA). CHO cell cultures were tested routinely for mycoplasma contamination. Human FPR (*hFPR1*, *hFPR2* and *hFPR3*) sequences were amplified by PCR and cloned into the Gateway entry vector, pDONR202, using the BP clonase enzyme mix, according to the manufacturers' instructions (refer to Supplementary Table 5 for primer sequences). The *hFPR* pDONR201 constructs were subsequently transfected into the pEF5/FRT/V5/dest vector using the LR clonase enzyme mix (Invitrogen Inc.). Receptor constructs (pEF5/FRT/V5/dest) were used to transfect FIp-In CHO cells (Invitrogen Inc.)[32]. Cells were selected using 500 μg per ml hygromycin to generate cell lines stably expressing each human receptor construct. Cells were maintained and cultured in high-glucose DMEM supplemented with 10% fetal bovine serum (FBS, from JRH Biosciences, Lenexa, KS, USA) and 500 μg per ml hygromycin B (Invitrogen Inc.) at 37 °C under a humidified atmosphere containing 5% $CO_2$. For intracellular calcium mobilization, extracellular signal-regulated kinase 1 and 2 phosphorylation (pERK1/2) and Akt1/2/3 phosphorylation, cells were seeded into 96-well transparent cell culture plates at $4 \times 10^4$ cells per well. After 6–8 h, cells were washed in serum-free DMEM and cultured overnight in serum-free DMEM at 37 °C in 5% $CO_2$. For cAMP accumulation, cells were seeded into 96-well transparent cell culture plates at $2 \times 10^4$ cells per well and grown overnight at 37 °C in 5% $CO_2$. Intracellular $Ca^{2+}$ mobilization, pERK1/2 and Akt1/2/3 phosphorylation (both Ser473 and Thr308) and inhibition of cAMP accumulation stimulated by Cmpd17b, Cmpd43 and the widely studied FPR-subtype-selective agonists fMLP (FPR1-selective agonist, from Sigma-Aldrich) was assessed in hFPR1-CHO and hFPR2-CHO cells.

*Intracellular $Ca^{2+}$ mobilization.* Intracellular $Ca^{2+}$ mobilization was determined in stably transfected recombinant hFPR-CHO cells[32]. Briefly, cells were washed once using HEPES-buffered saline (HBS) solution (150 mM NaCl, 2.2 mM $CaCl_2$, 2.6 mM KCl, 1.2 mM $MgCl_2$, 10 mM HEPES, 10 mM D-glucose; pH 7.4), and then incubated in Fluo-4-AM (1 μM; Invitrogen Inc.) in HBS containing 0.5% (w:v) BSA and 4 mM probenecid at pH7.4 for 60 min at 37 °C in 5% $CO_2$ in the dark[33]. Cells were then washed and placed in HBS/BSA/probenecid solution and assayed using the FlexStation 3 plate reader (Molecular Devices, Sunnyvale, CA)[32]. For all experiments, the peak change in fluorescence signal was normalized to the cellular response to 100 μM ATP, which was used as an internal positive control.

*Measurement of ERK1/2 Akt1/2/3 (Ser473) and Akt1 (Thr308) phosphorylation.* Surefire ERK1/2 and Akt1/2/3 phosphorylation kits were purchased from

PerkinElmer Life and Analytical Science (Waltham, MA, USA). Time course and concentration-response curves for pERK1/2 and pAkt1/2/3 were performed in serum-free DMEM at 37 °C in 5% $CO_2$. Concentration-response curves were generated at the peak-response time point as determined from time course experiments (5 min for fMLP and Cmpd43, 7 min for Cmpd17b). FBS (10%; v:v) was used as a positive control, with vehicle controls also included. Reaction termination followed by pERK1/2 and pAkt1/2/3 detection was performed[32,33]. Data were normalized to vehicle and the response elicited by 10% FBS.

*cAMP accumulation.* Cell culture medium was removed and cells incubated in stimulation buffer (1.4 M NaCl, 50 mM KCl, 8 mM $MgSO_4$, 2 mM $Na_2HPO_4$, 4.4 mM $KH_2PO_4$, 0.1% (w/v) BSA, 5 mM HEPES, 1.3 mM $CaCl_2$, 5.6 mM D-glucose; pH 7.4) with 10 mM rolipram (selective phosphodiesterase-4 inhibitor) at 37 °C for 30 min. Cells were then exposed to forskolin (3 μM) in the absence or presence of agonist and incubated for an additional 30 min at 37 °C. The reaction was terminated and cAMP detected[34]. Data were normalized to vehicle and the response elicited by 10 μM forskolin.

*Quantification of FPR agonist potency and maximal agonist response.* Nonlinear regression was performed using Prism 6.0 (GraphPad Software, San Diego, CA, USA). Concentration-response curves mediated by each agonist across the five signalling pathways (intracellular $Ca^{2+}$ mobilization, pERK1/2 phosphorylation, Akt1/2/3(Ser473) phosphorylation, Akt1(Thr308) phosphorylation and inhibition of cAMP accumulation) were fitted to derive the maximal agonist effect ($E_{max}$) normalized to the indicated positive control, and the ligand potency ($pEC_{50}$), defined as the negative logarithm of the agonist concentration that gives a response halfway between the lower and upper asymptotes of the concentration-response curve.

*Quantification of biased agonism.* Development of a quantitative framework of GPCR agonism allows ligand bias to be quantified and statistically analysed (Supplementary Fig. 8)[24,35]. A 'transduction coefficient', $Log(\tau/K_A)$ value, which reflects both agonist affinity ($K_A$) and efficacy ($\tau$) for a particular pathway, was derived for each agonist down each signalling pathway by refitting the concentration-response data to an extension of the Black–Leff operational model (see Supplementary Fig. 8). 'Bias factors' ($\Delta\Delta(\tau/K_A)$) were then calculated by normalizing the transduction coefficient ($Log(\tau/K_A)$) to a reference ligand, in this case Cmpd43, to generate $\Delta Log(\tau/K_A)$ values. This approach nullifies the influence of the natural system bias on the observed agonism and facilitates a quantitative cross-pathway comparison for each agonist. The Log(bias factor) for any one pathway over another was then estimated by normalizing the $\Delta Log(\tau/K_A)$ value to

a reference pathway, in this case Akt1/2/3(Thr308) phosphorylation. Using this approach, if the test agonist has no bias relative to the reference agonist, the Log(bias factor) should be 0.0 (that is, a bias factor of 1.0), irrespective of differences in response amplification between pathways. In contrast, significant deviation of Log(bias factor) away from 0.0 indicates the involvement of distinct conformations for the different agonists. To account for the propagation of error associated with normalizing the data to the reference pathway, statistical analysis was performed on the $\Delta Log(\tau/K_A)$ values.

All results for CHO cell signalling were compared with the widely studied FPR1-selective agonist fMLP[13,36]. All experiments in stably transfected CHO cells were performed in at least three separate experiments, each in triplicate, based on sample sizes routinely used for determination of cell signalling fingerprints[32–34].

**Animals.** All animal research was conducted in accordance with the National Health and Medical Research of Australian guidelines, and the approval was obtained from the Alfred Medical Research Education Precinct (AMREP) Animal Ethics Committee. Neonatal (1–2 day old) Sprague–Dawley rats and mice (both of mixed sex), and adult male C57BL/6 mice (11–14 weeks of age) were bred and housed in the AMREP Animal Centre and maintained under a 12 h light/dark cycle. All reagents were purchased from Sigma-Aldrich (St Louis, USA) except where indicated, and were of analytic grade or higher. The primer sequences used for this study is listed in Supplementary Table 4.

**Isolation and culture of primary cardiomyocytes and cardiac fibroblasts.** All materials used for cell isolation were tissue culture grade. Isolation of primary neonatal rat cardiomyocytes (NRCM) was performed by serial enzymatic digestion[37,38]. Neonatal rats and hearts were immediately placed in HBSS, 9.5 g per l Hanks Balanced Salts, 350 mg per l sodium bicarbonate. Ventricles were removed and digested in trypsin (1 mg per ml trypsin in HBSS) overnight at 4 °C and collagenase (2.5 mg per heart) four times at 37 °C for 10 min. Cardiomyocytes were suspended in sterile DMEM, supplemented with penicillin 100 U ml$^{-1}$, streptomycin 100 µg ml$^{-1}$ and 10% FBS. The cells were pre-plated twice (45 min at 37 °C) to reduce fibroblast contamination. Neonatal mouse cardiac fibroblast (NMCF) were obtained as a by-product of cardiomyocyte isolation. Isolation of primary neonatal mouse cardiomyocytes (NMCMs) followed the same protocol as that described for NRCM[37,38], with minor modifications. These modifications comprised using a lower concentration of trypsin (0.75 mg ml$^{-1}$) for the initial overnight heart incubation, a lower concentration collagenase type 2 (5.4 mg in 30 ml for the four serial digestion steps), a slower centrifugation step subsequent to serial digestion (350 g) and a single pre-plating step for cardiomyocyte enrichment (50 min at 37 °C).

NRCMs were plated on either 12-well tissue culture plates (for simulated ischaemia–reperfusion via H–R), 60 mm tissue culture dishes (for gene expression) or 96-well tissue culture plates (for determination of intracellular $Ca^{2+}$ mobilization), at a density of $1.3 \times 10^5$ cells per cm$^2$, in the presence of 1% 5-bromo-2′-deoxyuridine. NRCM intracellular $Ca^{2+}$ mobilization was determined as described for recombinant hFPR-CHO cells above (50 mM KCl was added to stop the cardiomyocytes from beating). NMCMs were plated on 12-well tissue culture plates, pre-coated with human fibronectin (1.25 µg cm$^{-2}$, BD Biosciences, Le Pont de Claix, France) at a density of $1.25 \times 10^5$ cells per cm$^2$, in the presence of 100 µM 5-bromo-2′-deoxyuridine. NMCFs (passage #3) were seeded on 60-mm tissue culture dishes at $9.5 \times 10^4$ cells per cm$^2$ and allowed to grow to 80% confluence before study.

Following 48 h in serum-free DMEM, NRCM in 12-well tissue plates were subjected to simulated ischaemia, induced by replacement of culture media with sterile-filtered Kreb's buffer (118 mM NaCl, 4.8 mM KCl, 1.2 mM $KH_2PO_4$, 1.2 mM $MgSO_4$, 25 mM $NaHCO_3$, 50 mM EDTA, 11.0 mM glucose, 1.75 mM $CaCl_2$) before incubation for 6 h at 37 °C under hypoxia (95% $N_2$–5% $CO_2$, using a hypoxia chamber, QNA International, Melbourne, Australia). At the end of hypoxia, Kreb's buffer was replaced with fresh, sterile-filtered Kreb's buffer, in the presence or absence of FPR agonists Cmpd17b, Cmpd43 (both 1 µM) or vehicle, and NRCMs were subjected to 48 h reoxygenation at 37 °C. At the end of 48 h reoxygenation, NRCM supernatant was collected for assessment of cardiomyocyte injury. Cardiomyocyte release of cTnI was determined as the marker of cell injury using a commercially available, high-sensitivity rat cTnI ELISA Kit (Life Diagnostic Inc.) as per the manufacturer's instructions. Separate NRVMs on 60-mm dishes were incubated for 48 h at 37 °C in serum-free DMEM, in the presence or absence of FPR agonists Cmpd17b, Cmpd43 (both 1 µM) or vehicle. At the end of 48 h, RNA was extracted and real-time PCR performed[39]) to determine the relative gene expression of rat Fprs (rFpr1, rFpr2 and rFpr3, refer to primer sequence in Supplementary Table 4) expressed as the threshold cycle number Ct subtracted from the maximum cycle number utilized, 40). Note that IUPHAR receptor nomenclature uses uppercase letters for human FPRs (hence FPR1 and so on), whereas title case is used for rodent receptors (hence Fpr1 and so on)[13].

Following 24 h in serum-free DMEM (supplemented with streptomycin, penicillin, insulin–transferrin–selenium premix (Gibco Laboratories, Grand Island, NY) and 0.5% BSA (Sigma-Aldrich)), NMCM in 12-well tissue plates were subjected to simulated ischaemia for 1.5 h at 37 °C, induced by replacement of culture media with a sterile-filtered metabolic inhibition buffer, with concomitant hypoxia (95% $N_2$–5% $CO_2$). The metabolic inhibition buffer comprised a modified

HEPES buffer, supplemented with 10 mM 2-deoxy-D-glucose and 20 mM D,L-lactic acid, pH adjusted to 6.5 (refs 7,8). Paired control NMCMs were incubated for 1.5 h at 37 °C in HEPES buffer. At the end of 1.5 simulated ischaemia or sham, NMCM buffer was replaced with sterile-filtered Kreb's buffer, in the presence or absence of FPR agonists Cmpd17b, Cmpd43 (both 1 µM) or vehicle, and NMCMs were subjected to 2.5 h reoxygenation at 37 °C, before collection of NMCM supernatant for assessment of cardiomyocyte injury via cardiomyocyte release of cTnI, using a mouse cTnI ELIZA Kit (Life Diagnostic).

NMCFs were starved overnight in serum-free DMEM before incubation with the pro-fibrotic stimulus TGF-β (10 ng ml$^{-1}$) for 24 h at 37 °C. FPR agonists Cmpd17b, Cmpd43 (both 10 µM) or vehicle were present 30 min before TGF-β. At the end of 24 h stimulation, RNA was extracted and real-time PCR performed to determine the relative gene expression of Ctgf and IL-1β, relative to housekeeping gene 18S (ref 39). Relative gene expression of mouse Fprs (mFpr1, mFpr2 and mFpr3, refer to primer sequence in Supplementary Table 4) expressed as the threshold cycle number Ct subtracted from the maximum cycle number utilized, 40 was also determined at the end of 24 h stimulation. All experiments in primary cardiac cells were performed in at least four separate experiments, in paired comparisons with untreated cells from the same cardiac cell preparation, where each separate cardiac cell preparation represents a single n (refs 38,39).

**Cardiac injury responses in vivo.** Adult C57BL/6 mice were randomly assigned to myocardial ischaemia or sham in vivo. The impact of small-molecule FPR agonists Cmpd17b and Cmpd43 on myocardial I–R injury was then assessed in three separate cohorts of C57BL/6 mice subjected to I–R, with a fourth cohort subjected to MI via 4 weeks' permanent LAD occlusion. For assessment of all cardiac injury responses in mice in vivo, sample size per group was chosen based on sample sizes routinely used for myocardial ischaemic studies in mice[25,40]. All analyses undertaken in mice in vivo were performed blinded to the investigator, including IS determination, histology, immunofluorescence and cardiac function analysis. All animals studied in vivo were included in all analyses, unless exhibiting data more than two s.d.'s from the mean, identified as outliers by the GraphPad Prism. Anaesthetized mice (ketamine 80 mg kg$^{-1}$, xylazine 20 mg kg$^{-1}$ and atropine 1.2 mg kg$^{-1}$, KXA, i.p.) underwent reversible LAD ligation[25]. Following sedation, mice were intubated and ventilated (Harvard Apparatus, MA, USA) with room air mixed with oxygen (tidal volume 0.25 ml, 150 breaths per min) and placed on a heating pad. A left thoracotomy was to be made around the third intercostal space, where the beating heart was located. The LAD was reversibly ligated using 7-0 silk suture with a slipknot enclosing two releasing rings. Regional ischaemia was confirmed by pale colour of the ligated area. Air was then evacuated from the chest, the cavity closed and normal respiration restored. Blood flow through the left coronary artery was subsequently re-established at the end of the ischaemic period (that is, reperfusion) by releasing the slipknot[25,42].

Mouse cohorts 1 and 2 were subjected to 40 min ischaemia with either 24 h or 7-day reperfusion, optimal time points for the assessment of cardiac necrosis and early cardiac remodelling in vivo, respectively, with three separate batches of mice studied per cohort (each comprising sham, vehicle-, Cmpd17b- or Cmpd43-treated I–R mice). Cohort 3 was subjected to 60 min LAD occlusion with 48 h reperfusion, for the quantification of cardiac and systemic inflammation with four separate batches of mice studied per cohort (each comprising sham, vehicle-, Cmpd17b- or Cmpd43-treated I–R mice). Sham-operated mice underwent identical surgical procedures, except the LAD was not ligated. Mice were randomly assigned to administration of FPR agonists, either Cmpd17b or Cmpd43 (both 50 mg kg$^{-1}$ per day i.p.), or equivalent volume of vehicle control (10% dimethylsulphoxide containing 0.8% Tween-20 in saline i.p.), with the first dose immediately before reperfusion (and daily thereafter). FPR agonist doses in vivo were chosen based on that previously shown for Cmpd43 to reduce ear inflammation[22], with pilot studies suggesting plasma concentrations in the high submicromolar–low micromolar range for both compounds for several hours post dose.

An additional fourth cohort of mice, Cohort 4, was subjected to a more prolonged ischaemic insult, permanent occlusion of the LAD; late cardiac remodelling and cardiac dysfunction were assessed after 4 weeks in sham and MI mice ± vehicle or FPR agonist (50 mg kg$^{-1}$ per day i.p.) at time of cardiac surgery (with four separate batches of mice studied per cohort (each comprising sham, vehicle- or FPR agonist-treated MI mice). For all cohorts, infarcted or sham-operated mice were killed under KXA anaesthesia at study end, and heparinized blood collected by cardiac puncture for later analysis as indicated.

**Assessment of cardiac necrosis in vivo.** The optimal time point for assessment of cardiac necrosis is 24 h reperfusion after an ischaemic insult[25,40]. Plasma levels of cTnI were determined at this time point first using a commercially available, high-sensitivity mouse cTnI ELISA Kit (Life Diagnostic Inc., Pennsylvania, USA) as per the manufacturer's instructions. To further evaluate the extent of cardiac necrosis, IS in relation to the AAR was also determined using 2,3,5-triphenyltetra-zolium chloride staining. The LAD was tightly re-occluded in anaesthetized mice after 24 h reperfusion and Evans blue dye (0.1 ml, 5%) was injected as a bolus into the LV. The heart was then excised and rinsed in cold saline to remove excess dye. The LV was isolated, frozen at −20 °C and then cut transversely into six to seven slices at 1.0 mm thickness. LV slices were incubated for 45 min with 1.5% 2,3,5-triphenyltetra-zolium chloride solution at 37 °C. The presence of Evans blue

indicated perfusion, whereas its absence indicated lack of perfusion to that region. Brick red areas indicated viable myocardium, while white or yellowish regions demarcated necrotic tissue. The slices were mounted between glass slides, and images were acquired digitally using a surgical microscope (Leica Wild M3B, Heerbrugg, Switzerland) coupled with digital camera (Nikon Cool-PIX4500, Tokyo, Japan). The images were analysed using the Image J analysis programme (Version 1.45S, National Institute of Health, USA). The non-ischaemic zone (blue area), AAR zone (red and white or yellow areas), infarct zone (white or yellow areas) and total LV were outlined and quantified blindly. IS was calculated as the percentage of infarct zone in the AAR[25,40].

**Assessment of cardiac remodelling in vivo.** LV tissues collected from mice in Cohort 2, after 40 min ischaemia and 7-day reperfusion (optimal time point for assessment of cardiac fibrosis and apoptosis), were fixed in neutral buffered formalin, embedded in paraffin by the Alfred Pathology Service (Melbourne, Australia) and sectioned at 4 µm with a Leica 2135 microtome (Leica Microsystems, Wetzlar, Germany). Sections were stained with picrosirius red (0.1%, Fluka, Bucks, Switzerland; pH 2) for assessment of cardiac collagen deposition. Images were taken at × 1.25 to capture the whole LV. To examine the peri-infarct area, images were collected at × 20 using the microscope (Olympus BX61, Olympus Inc., ON, Canada) and QCapture Pro software (version 5.1 for Windows, Media Cybernetic Inc., MD, USA). The area of picrosirius red staining (% LV area) was quantitatively measured using the Image-Pro Plus software (Media Cybernetic Inc.) for sets of LV slides from the 7-day reperfusion study. Levels of cell death within the infarct area were also assessed in paraffin-embedded ventricular sections, using the CardioTAC In Situ Apoptosis Detection Kit (Trevigen, Gaithersburg, MD, USA)[39]. This method detects nuclear DNA fragmentation by using a terminal deoxynucleotidyl transferase enzyme, which incorporates labelled nucleotides on the free 3′-OH ends of DNA fragments. Positively stained apoptotic cells were distinguished by blue staining, while negatively stained cells were counterstained red with Nuclear Fast Red. The ratio of dead:alive cells was quantified, and the results expressed as fold levels detected in sham mouse heart.

**Assessment of cardiac inflammation in vivo.** Lungs, atria, left and right ventricles from mice in Cohort 3 (following 60 min ischaemia and 48 h reperfusion, optimal timing for detecting cardiac leukocyte infiltration) were dissected, blotted dry and weighed. LVs were cut into half at the occlusion site, and the apical half placed in Tissue-Tek optimal cutting temperature compound (Tissue-Tek, Torrance, USA) for storage at − 80 °C. LV tissues were then sectioned at 6 µm for immunofluorescent detection of cardiac macrophage and neutrophil content. Sections were pre-incubated with 4% paraformaldehyde for 20 min and 10% normal goat serum for a further 30 min. Sections were then incubated at room temperature with either CD68[+] primary antibody or Ly-6B.2 primary antibody (1:200, ABD Serotec, Raleigh, USA) for 1 h, followed by 30 min incubation with the Alexa Fluor 546 secondary antibody (1:200, Invitrogen Inc.). Finally, sections were incubated with 0.001% Hoechst 33342 (Invitrogen, Melbourne, Australia) for 30 min to elicit nuclear staining. Single images were photographed using a Nikon A1R confocal microscope (Nikon Instruments Inc., NY, USA) under × 20 magnification, and 9 × 9 single images were then automatically stitched together using the NIS-Elements AR software (version 4.10 for Windows, Nikon Instruments Inc.) to form each complete composite LV image. The images for infarct area were captured under × 40 magnification, and the stitched images were then analysed using the Fiji software (version 1.48c for Max OS X, National Institutes of Health, USA). The threshold was set for each image to enable optimal sensing and quantification of red fluorescence intensity above the threshold, and the software then automatically quantified these red fluorescent signals.

LV tissues were also collected from anaesthetized mice in Cohort 4 at study end point, 4 weeks post MI. LV RNA was extracted from the infarct and non-infarct AAR, and gene expression determined via real-time PCR using SYBR Green chemistry (Applied Biosystems, Scoresby, Victoria, Australia). Primers were generated from murine sequences published on GenBank (refer to Supplementary Table 4 for primer sequence). Quantitative real-time analysis was performed using ABI Prism 7700 Sequence Detection and the ΔΔCt method was employed to obtain relative fold differences in expression[39].

**Assessment of systemic inflammation in vivo.** Whole blood and plasma were also collected from mice in Cohort 3 (following 60 min ischaemia and 48 h reperfusion) to assess total and differential circulating WBC numbers, and plasma levels of the pro-inflammatory cytokine IL-1β. Blood samples were diluted 1:20 into 2% acetic acid to lyse the red blood cells for total WBC counting, using a hemocytometer. Blood smears were stained using a Hemacolor staining kit (Merck Millipore, Melbourne, Australia) and then examined under × 60 magnification using an Olympus Biological CHS microscope (Olympus Inc., Tokyo, Japan) for manual counting of neutrophils, lymphocytes and monocytes per 100 WBCs. Plasma IL-1β was determined using a commercially available ELISA Kit (elizakit, Melbourne, Vic, Australia) as per the manufacturer's instructions.

**Quantitation of FPR agonist plasma concentrations in vivo.** Plasma samples collected from mouse Cohorts 1 and 3 (after 24 and 48 h reperfusion, respectively),

~20 h post-final dose, were analysed for plasma FPR agonist concentrations by the Monash University Centre for Drug Candidate Optimisation (Parkville, VIC, Australia). The FPR agonists were extracted from plasma samples using acetonitrile precipitation (2:1) and analysis was conducted using a Waters Xevo TQ triple quadrupole mass spectrometer coupled to a Waters Acquity UPLC (Waters Corporation, Milford, MA). Mass spectrometry was performed in positive-mode electrospray ionization using multiple reaction monitoring, with MS–MS transitions of 456.01 > 285.07 and 385.11 > 231.96 for Cmpd17b and Cmpd43, respectively. UPLC conditions comprised a Supelco Ascentis Express RP Amide column (50 × 2.1 mm, 2.7 µm), with an acetonitrile–water–0.05% formic acid gradient mobile phase, gradient cycle time of 4 min and flow rate 0.4 ml min$^{-1}$. The elution times of Cmpd17b and Cmpd43 were 2.24 and 1.66 min, respectively, while that of the internal standard diazepam was 1.79 min. The lower limit of detection for both FPR agonists was 1 ng ml$^{-1}$.

**Echocardiography.** M-mode echocardiography was performed in anaesthetized mice allocated to Cohort 4 (ketamine/xylazine/0atropine: 60/6/0.7 mg kg$^{-1}$ i.p.) to obtain measures of LV function, at baseline and after 1 and 4 weeks post MI, utilizing a Philips iE33 ultrasound machine (North Ryde, NSW, Australia) with a 15 MHz linear (M-mode) transducer, as previously described[40]. LV posterior wall thickness, LV chamber dimensions and fractional shortening were assessed from M-mode echocardiography[39].

**Data analysis.** All data were analysed using GraphPad Prism 6.0 (GraphPad Software). Results are expressed as mean ± s.e.m. unless otherwise stated. Statistical analyses utilized Student's t-test or one-way ANOVA followed by Dunnett's or two-way ANOVA followed by Tukey's comparison post hoc test, as appropriate. Values of $P < 0.05$ were considered statistically significant.

**Data availability.** The data that support the findings of this study are available from the corresponding author on request.

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

## Acknowledgements

We acknowledge the Monash Micro-Imaging facility (provision of instrumentation and training), and the Monash Centre for Drug Candidate Optimization (bioanalysis support). This work was supported in part by both the National Health and Medical Research Council (NHMRC) of Australia, including APP1045140 (to R.H.R., X.-M.G. and Y.H.Y.), APP1055134 (to P.M.S. and A.C.) and the Victorian Government's Operational Infrastructure Support Program. R.H.R. and X.-J.D. are NHMRC Senior Research Fellows (APP1059960 and APP1043026, respectively). L.T.M. is an ARC DECRA Postdoctoral Research Fellow (DE130100117); A.C. and P.M.S. are NHMRC Principal Research Fellows (APP1041875 and APP1059015, respectively).

## Author contributions

R.H.R., C.X.Q., L.T.M., X.-M.G. study conception and design; C.X.Q., L.T.M., R.L., N.C., S.R., M.D., A.E.A., D.H., X.-J.D., X.-M.G. and R.H.R. performed experiments; C.X.Q., L.T.M. and R.H.R. analysed data; C.X.Q., L.T.M., J.E.B., Y.H.Y., A.S.G., D.M.K., X.-J.D., P.M.S., A.C., X.-M.G. and R.H.R. interpreted results; C.X.Q. and R.H.R. prepared figures; C.X.Q., L.T.M., X.-J.D., P.M.S., A.C., X.-M.G. and R.H.R. drafted manuscript; C.X.Q., L.T.M., X.-J.D., P.M.S., A.C. and R.H.R. edited and revised manuscript.

## Additional information

**Competing financial interests:** The authors declare no competing financial interests.

**Publisher's note**: 

