## [Peer Review File · Nature Communications]

Reviewers' comments:

Reviewer #1 (expert in MI therapeutics)

Remarks to the Author:

In this manuscript Qin et al present the novel observation that a biased agonist at FPR1/FPR2 is protective, at variance from a 'classical' non biased compound. The study is well planned, executed and presented. I have found of particular interest at the Authors provide a way forward to 'translate' the complex biology of FPRs towards definition of what is necessary to guide the development of novel therapeutic agents developed on this intriguing family of receptors. As such, the study fits perfectly in what proposed in a recent review (which should be quoted; PMID: 26478210) - the need to identify and exploit the cell specific signalling that is predictive to tissue protective activity.

I have a few comments:

1. In the Abstract the Authors should define what compound 43 is, in the sense of telling the reader that is not a biased agonist and binds both FPR1 and FPR2.
2. Lines 116 versus 122: is fMLP a stimulator or an inhibitor of cAMP?
3. Line 155 - describe what is the impact of cellular Fpr1/2 expression the Authors are referring too. Is it an induction?
4. Lines 209-211 - I agree. the Ref 29 (which is quoted later) could be quoted here.
5. Line 245 - I would remove potentially. 'May' is enough to afford uncertainty.
6. Figure 3 - being an in-vitro study this experimental protocol allows to generate full concentration-responses thus defining whether the differences are just a matter of potency or truly linked to the biased VERSUS non-biased agonism. This is particular important as this point could also be addressed through in vivo protocols, though clearly more expensive and time-consuming. I would concentrate on the 24h response for fibrotic factors.

Reviewer #2 (expert in FPR agonists)

Remarks to the Author:

A. Summary of the key results

The paper by Qin et al is based targeting of the formyl peptide receptors (FPRs) in myocardial infarction (MI). Specifically, the authors focus on identifying that FPR1/FPR2-biased agonism (i.e. the ability of GPCRs to adopt different conformational states, each linked to distinct cellular outcomes) may represent a novel therapeutic strategy for the treatment of this disease, and have suggested that small molecule FPR agonists with improved cardioprotective profile may be of benefit for treating myocardial infarction (MI).

B. Originality and interest: if not novel, please give references

This is an interesting paper, although lacks attention to detail in parts. Further experiments are needed in order to confirm the conclusions that have been drawn. If not, then it is not possible to draw the conclusions that authors state. Also, in a physiological setting, where only one dose is used, effects observed could simply be related to dose, rather than bias.

C. Data & methodology: validity of approach, quality of data, quality of presentation

Overall, as discussed below, several aspects of the data and methodology need to be fixed e.g. vehicle controls and inclusion of other FPR1 and FPR2 specific agonists needs to be added to the study before the conclusion that FPR1/2 biased agonism represents a novel strategy to limit MI. Data presented also needs attention to detail, as discussed in the minor points below.

D. Appropriate use of statistics and treatment of uncertainties

Tests are fine.

E. Conclusions: robustness, validity, reliability

As discussed, without changes to the experimental design, it is not really possible to draw the conclusions that authors state.

F. Suggested improvements: experiments, data for possible revision

Major concerns

- 1) Whilst the authors have stated that CHO cells do not express native Fprs (please provide reference), it would have been more relevant to use a human cell line e.g. HEK cells. It would be of benefit to repeat the key findings in HEK cells stably transfected with FPR1 or FPR2 as per the work of J. Gao and P. Murphy.
- 2) Whilst it is true that fMLP is a high affinity FPR1-selective peptide agonist, it does activate FPR2 at high concentrations. Therefore, it is critical to perform these experiments with an FPR1 specific agonist (note change needed on page 15 (see minor point 4 below). Also, add the data with the FPR2 specific agonist LXA4.
- 3) Is only FPR1 expressed on cardiomyocytes? What about Fpr2? It is unclear from the text.
- 4) Supplementary figure 2, an n=1 is insufficient to draw conclusions. Further n numbers are needed.
- 5) It was interesting to note that LXA4 failed to stimulate ERK-phosphorylation. - More discussion is needed here.
- 6) No vehicle was included in the figures, and needs to be included. This is important as especially as the vehicle for LXA4 is usually ethanol, and ethanol has been shown to induce e.g. ERK activation on its own (Christophe et al., 2002). Also, it would be important to confirm findings using a drug that does not signal through FPRs. Include the data.
- 7) Why were rat cardiomyocytes and mouse cardiac fibroblasts chosen, rather than both from the same species? As the in vivo methods were performed in mice it would be worth validating the findings in mouse cardiomyocytes, rather than rat, or perform the in vivo work in rats, not mice.
- 8) Results. Page 9 + Figure 4. Why was the dose 50mg/kg chosen for both compounds? Are the effects dose dependent? Include further doses. Also what happened with fMLP and LXA4 in these studies? Provide this data to strengthen the paper.
- 9) Results. Page 9 + Supplementary Figure 4. These figures are poor. It is impossible to see whatever it is the arrows are pointing to. Neutrophils I assume?
- 10) What are the circulating counts for mice treated with these compounds? - Provide.

- 11) n numbers throughout are extremely variable, especially in vivo (ranging from n=4-14). What is the reason for this?
- 12) Figure 3. Variability in results. Increase n numbers to 7 or 8, not 4.
- 13) Do the compounds fMLP and LXA4 have any effect on differential circulating blood counts? Please provide all this data.
- 14) Although plasma Il-1beta levels were measured, what about effects on other pro- and anti-inflammatory cytokines. Were any of these altered? Provide data.
- 15) n=5 for cells counts (I/R) was performed and yet n=8 for Il-1beta measurement. As the data is variable, it would be worth adding some more animals to the I/R group, especially as effects on circulating neutrophils and monocytes may be then seen.
- 16) Page 10 first paragraph states that "the I-R-induced increases in total WBC were and neutrophils were absent in Cmpd17b treated mice. Figure 6 shows a trend for these absent responses with Cmpd17b, but is not significant. Re-phrase this incorrect statement on page 10.
- 17) Page 10 states that "taken together our observations suggest in contrast to Cmpd43, Cmpd17b attenuates the early inflammatory response associated with reperfusion injury". Could these effects simply be dose dependent?
- 18) What happens at 24 h and 48 h on cardiac apoptosis. Do the compounds effect apoptosis at these time points?
- 19) There are FPR1 and FPR2 knock out animals. What happens with the compounds in these mice? It would strengthen the manuscript to include at least one.
- 20) Figure 6 shows circulating WBC and differential counts and plasma il-1b following 48 h reperfusion. What happens at 24h and 7 days? Include this data.
- 21) Figure 7. Images a+c. It would be helpful to include pictures at a higher magnification, as it is hard to see anything.
- 22) Figure 7. Panel b: why n=4? It would be worth increasing these n numbers to equate with e.g. panel d. Also, sham in panel b looks as those there are only 3 values, not 4 as stated. Panel b.
- 23) Figure 7. Is Cmpd17b not significant to sham, especially as I/R is?
- 24) Conclusions. Whilst this paper clearly focuses on myocardial models, very little discussion is provided on effects on other cells of these compounds, particularly in the in vivo setting. FPRs are expressed on a number of myeloid cells and endothelium. Discussion of the role that the different cell types play should be discussed and not excluded.

Minor

- 1) Introduction (p5). More information regarding Cmpd43 and Cmpd17b should be provided e.g. Specific FPRs they bind to.
- 2) FPR2 was previously known as 'FPR-like 1' not 'FPR-like' R. Amend accordingly
- 3) Introduction (p6) - remove the word 'hence' from paragraph two.

4) Page 15. fMLP is not FPR1 selective. It binds to FPR2 at lower affinity. This is misleading and should be amended.

5) Supplementary Figure 4. What are the arrows for? No mention in the figure legend.

6) Figure 4a. These images are of poor quality. It would be worthwhile to include all slices taken for each heart.

7) Supplemental Figure 2D - Include line connecting dots as per other graphs.

8) Be consistent in the figures and manuscript. Either use capitals for each or lower-case i.e. A or a, but not a combination.

9) Figure 2 has two d's i.e. cAMP should be image e, not d.

10) Page 8, first paragraph, remove comma after GPCR.

11) All figure legends should have doses/concentrations of the drugs used.

12) Page 10 second paragraph, Cmpd17b is missing the 'b'.

13) Supplemental. Where are Cmpd17b and Cmpd43 obtained from?

14) Figure 5 a+c. These figures are unclear, either clearly emphasise the point (e.g. add magnified images) or remove.

G. References: appropriate credit to previous work?

References are fine.

H. Clarity and context: lucidity of abstract/summary, appropriateness of abstract, introduction and conclusions

These are fine.

Reviewer #3 (expert in biased agonists and cardiac function)

Remarks to the Author:

A. Summary of key results:

In this study, Qin et al. report that the FPR1/2 agonist Cmpd17b is biased against the induction of Ca²⁺ signaling, while retaining P-ERK and P-Akt activation, versus the conventional agonist Cmpd43, suggesting that it could promote survival signaling in the absence of Ca²⁺-mediated cell death and inflammation. Indeed, 17b reduces H-R-induced cardiomyocyte toxicity as well as TGFβ-induced inflammatory gene expression in cardiac fibroblasts in vitro. Further, 17b was shown to reduce the injury and immune responses to I-R in vivo leading to altered cardiac remodeling.

B. Originality and interest:

Overall the study offers exciting new insight into the potential therapeutic utility of biased ligands, particularly for FPR, in achieving better outcomes following cardiac injury.

C. Data and methodology:

Overall the methodology is performed, and data presented, well. However:

1. Peak cell death after ischemic injury occurs rapidly and subsides over time, thus it's unclear why the authors measured TUNEL at 7 days post-I-R versus using their 24-hr timepoints when the bulk of the cell death will have occurred and larger differences in 17b-mediated protection would be observed.

D. Statistics:

1. Authors should define the statistical significance and test used for Figs 1F and 2F in the legends.
2. In the results section, referring to the data in Fig. 7D the authors state that "the impact of I-R on LV fibrosis in this region was absent in Cmpd17b-treated mice", which based on the statistics is not true.

E. Conclusions:

1. A significant study limitation is that the signal bias of 17b shown in exogenous cells has not been linked to its impact on in vitro and in vivo injury models. 17b is clearly more protective than cmpd43, but the proximal signaling responses in these models have not been investigated.
2. Some of the statements regarding the ability of 17b to block effects versus I-R condition in vivo are too strong considering the data does not appear statistically significant (fibrosis, immune cell levels).

F. Suggested Improvements:

1. Does the signal bias translate to primary myocytes and fibroblasts? While in theory it should, it is important to show that at least the difference in Ca²⁺ responsiveness with preserved P-ERK/Akt signaling is recapitulated in these cells, thereby linking the signal bias data in exogenous systems to the measured outcomes post-H-R and post-TGF β in the endogenous primary cell models of interest. The authors have previously reported that FPR1 in particular is responsible for the cardioprotective benefit of the FPR ligands ANX-A1 and Ac-ANX-A1(2-26) (Qin et al. 2013), however it would be of benefit to show in the isolated myocyte and fibroblast models that the effects of 17b via endogenous FPRs can indeed be blocked with an FPR antagonist.
2. One of the pieces of data for the I-R studies that is not shown is the plasma level of each of the compounds measured. The authors state in the discussion that they were measured 7 days after treatment in vivo and were not different, but in the supplemental methods state they were measured several hours after dosing. Are the compounds really still present after 7 days, and if so is this a relevant timepoint for comparison of bioavailabilities? Given the timeframe of acute cell death and inflammation that occurs immediately after I-R, it would make more sense to report the plasma levels of each compound around the time of surgery when enhanced survival signaling would be expected to dampen the initial cell death response. This information is critical and the values should be shown since differences in the injury and immune responses could result from different pharmacokinetics if the timing of intervention is essential. It's also critical to show empirically that Cpd43 was found at similar levels to 17b since, as the authors note, Cpd43 did not produce any significantly different effects from I-R alone.
3. Cardiac function is not shown, which is a major limitation: although 17b decreases cell death and

inflammation, the ultimate therapeutic goal is to preserve or improve cardiac contractility over time (several weeks post-I-R). This is also important to know based on the gravimetric data wherein the 17b-treated I-R hearts appear to become smaller than sham hearts after 7 days. What is causing this effect, and are the hearts functioning normally? Since there are no sham + compd controls, the effect of 17b in the context of I-R cannot be separated from any long-term impact on cardiac structure it may have in the absence of injury.

G. References: No concerns

H. Clarity and Context:

1. The authors do not state whether they think that the effects of 17b on immune cell infiltration into the heart (and circulating levels of immune cells) are due to the decreased injury itself, or direct effects on immune cells themselves. This interpretation could change the importance of the data. For instance the decreased infiltration and immune cell levels in blood are consistent with reduced injury and could be presented more clearly as such. However, if the authors think 17b impacts the immune cells themselves, then additional in vitro migration/chemotaxis assays could be performed to strengthen this idea since none of the blood cell data is statistically significant as reported (17b vs I-R).

“Small-Molecule Biased Formyl Peptide Receptor Agonist Compound-17b Protects Against Myocardial Ischemia-Reperfusion Injury in Mice”

Cheng Xue Qin, Lauren May, Renming Li, Nga Cao, Sarah Rosli, Minh Deo, Amy Alexander, Duncan Horlock, Jane Bourke, Yuan Yang, Alastair Stewart, David Kaye, Xiao-Jun Du, Patrick Sexton, Arthur Christopoulos, Xiao-Ming Gao, and Rebecca Ritchie.

RESPONSE TO REVIEWERS

We are encouraged by the positive response from the Reviewers to our manuscript, and are pleased that the potential pathological importance of biased agonism at formyl peptide receptors was recognized. The robust prevention of multiple aspects of myocardial injury associated with this biased agonism, across isolated cardiac cells (myocytes and fibroblasts) and excitingly, two distinct pre-clinical mouse models of myocardial injury *in vivo*, reveals potential new mechanistic insights for development of biased FPR agonists for as therapy for cardiac pathologies such as heart attack (myocardial infarction).

We thank the reviewers for the constructive comments and suggestions for improving our manuscript. We have endeavored to address all of the reviewers concerns and comments within the current revision, outlined below in our detailed Response to Reviewers and underlined in red within the manuscript. Considerable new data has been included to address all matters raised by reviewers and we believe that the quality of the revised manuscript has clearly benefited from these additions. We have added two new co-authors, Prof David Kaye and Mr Duncan Horlock, who have provided significant contribution with respect to the new mouse cardiomyocyte studies, in addition to intellectual assistance with respect to manuscript revision.

We have endeavored to ensure our revised manuscript complies with the editorial policies of *Nature Communications*, and have completed and uploaded the checklist for formatting requirements with the revised article, in addition to the completed checklist for reporting requirements. This has included transferring significant methodological details from the Supplement to the Main text Methods, to ensure sufficient Methods are contained within the main paper.

REVIEWER 1

Remarks to the Author.

In this manuscript Qin *et al* present the novel observation that a biased agonist at FPR1/FPR2 is protective, at variance from a 'classical' non biased compound. The study is well planned, executed and presented. I have found of particular interest at the Authors provide a way forward to 'translate' the complex biology of FPRs towards definition of what is necessary to guide the development of novel therapeutic agents developed on this intriguing family of receptors. As such, the study fits perfectly in what proposed in a recent review (which should be quoted; PMID: 26478210) - the need to identify and exploit the cell specific signalling that is predictive to tissue protective activity.

The review referred to by Reviewer 1 (Perretti M, Leroy X, Bland EJ, Montero-Melendez T. Resolution Pharmacology: Opportunities for Therapeutic Innovation in Inflammation. *Trends Pharmacol Sci* 2015; 36: 737-755), cited in the original submission (Ref#6) is now additionally cited several times in both the Introduction (page 5, paragraphs 2 and 3) and the Discussion (page 13, paragraph 2) of the revised manuscript, emphasizing that “FPRs are inherently dynamic, interacting with multiple, structurally-diverse naturally-occurring ligands that stimulate qualitatively-different physiological responses with respect to inflammation and its resolution, dependent on both the cell-type and the ligand”.

Comments:

1. In the Abstract the Authors should define what compound 43 is, in the sense of telling the reader that is not a biased agonist and binds both FPR1 and FPR2.

We have revised the abstract accordingly, “The small-molecule FPR1/FPR2 agonist, Compound-17b (Cmpd17b), exhibits a distinct signaling fingerprint to the conventional FPR1/FPR2 agonist, Compound-43 (Cmpd43)”.

2. Lines 116 versus 122: is fMLP a stimulator or an inhibitor of cAMP?

The cAMP responses to Cmpd43 and fMLP in hFPR1-CHO cells are bi-phasic and concentration-dependent. Lower concentrations of Cmpd43 and fMLP inhibit forskolin-stimulated cAMP accumulation, whereas higher concentrations stimulate cAMP accumulation. This has been reworded as follows on page 8 of the revised manuscript:

“However, a bi-phasic concentration-response curve was evident for the influence of Cmpd43 and fMLP on forskolin-stimulated cAMP accumulation, switching from an inhibitory to a stimulatory response at higher concentrations. Under these circumstances, potency (pEC_{50}) values were derived over the lower concentration range, reflecting agonist-mediated inhibition of cAMP accumulation.”

3. Line 155 - describe what is the impact of cellular Fpr1/2 expression the Authors are referring too. Is it an induction?

We have now clarified the cellular Fpr1/2 expression referred to in the Results section. This now reads (see page 10 of the revised manuscript) “*The relative gene expression of Fprs in cardiomyocytes is $rFpr1 > rFpr2 >> rFpr3$; this is similar to cardiofibroblasts, except $mFpr3$ is not detected (assessed via real-time PCR, Supplementary Fig. 3). The impact of treatment with FPR-agonists Cmpd17b and Cmpd43 on cellular Fpr1/2 gene expression was also determined in cardiomyocytes and cardiofibroblasts. Agonist-treatment elicited minimal impact on Fpr gene expression in either cardiac cell type, with the exception of the trend for Cmpd43 to reduce cardiomyocyte $rFpr2$ ($P=0.05$) and to blunt the TGF- β -mediated reduction in cardiofibroblast $mFpr1$ (Supplementary Fig. 3).”*

We have also expanded the description the data presented in Supplementary Figure 3 in the Supplementary Results.

Please note that we have endeavoured to remain consistent with the International Union of Basic and Clinical Pharmacology nomenclature for FPRs (Ref#13), which uses uppercase letters for human formyl peptide receptors (e.g. FPR1) whereas title case is used for rodent receptors (e.g. Fpr1).

4. Lines 209-211 - I agree. the Ref 29 (which is quoted later) could be quoted here.

We now also cite this reference (Ref #26 of the revised manuscript) on pages 13 and 16 of the revised manuscript, as suggested.

5. Line 245 - I would remove potentially. 'May' is enough to afford uncertainty.

We have revised the text as suggested, removing “potentially” on page 13 of the revised manuscript.

6. Figure 3 - being an in-vitro study this experimental protocol allows to generate full concentration-responses thus defining whether the differences are just a matter of potency or truly linked to the biased VERSUS non-biased agonism. This is particular important as this point could also be addressed through in vivo protocols, though clearly more expensive and time-consuming. I would concentrate on the 24h response for fibrotic factors.

We have now performed substantial new experiments to specifically address these comments. These include:

(i) **performing concentration-response studies to both FPR-agonists with respect to cardiomyocyte intracellular Ca^{2+}_i mobilization (Ca^{2+}_i)**, in neonatal rat cardiomyocytes. These data demonstrate that our conventional FPR-agonist Cmpd43 elicits concentration-dependent increases in Ca^{2+}_i ; in contrast, Cmpd17b elicits the opposite effect, yielding a concentration-dependent decrease in Ca^{2+}_i (new Figure 3c). This confirms that Cmpd17b elicits biased signalling in primary cardiomyocytes at the level of Ca^{2+}_i .

(ii) **determination of plasma levels of Cmpd17b and Cmpd43 after 24h and 48h reperfusion**. As illustrated in the revised manuscript (new Figure 5c), the plasma levels of Cmpd43 *in vivo* are ~5-fold those of Cmpd17b at both timepoints, yet Cmpd43 is a more potent agonist than Cmpd17b at each of the 5 different signals determined *in vitro*, on the kinase survival pathways, as well as Ca^{2+}_i (noting that the plasma levels of Cmpd43 *in vivo* exceeded EC_{50} in CHO cells *in vitro*).

Taken together, these data suggest that the comparative lack of cardioprotection evident in response to Cmpd43 *in vivo* is not due to an insufficient dose, but rather is more readily explained by differences at the level of biased signaling away from Ca^{2+}_i . In line with these new findings, we have now revised our Discussion. Please refer to the paragraph at the end of page 14 (continuing into page 15), which now reads: *“Based on their opposing effects on primary cardiomyocyte Ca^{2+}_i , and their relative plasma levels in vivo, an insufficient dose or lower bioavailability of Cmpd43 relative to Cmpd17b does not provide an alternative explanation for its lack of cardioprotective actions. Furthermore plasma levels of Cmpd43 were approximately 5-fold those of Cmpd17b, yet Cmpd43 is a more potent agonist than Cmpd17b on the kinase survival pathways, as well as Ca^{2+}_i in vitro. Collectively, our findings suggest that the differences in cardioprotection in vivo between Cmpd17b and Cmpd43 may be more readily explained by FPR biased agonism at the level of Ca^{2+}_i . Relative to Cmpd43, Cmpd17b activation of cardioprotective signaling is not countered by concomitant potent increases in Ca^{2+}_i (which in the case of Cmpd43 likely negate its potential confer cardioprotection).”*

We have not undertaken analyses of fibrotic factors after 24h *in vivo*, because this is likely to be too early a timepoint for fibrotic responses following I-R injury.

REVIEWER 2

Remarks to the Author.

A. Summary of the key results

The paper by Qin et al is based targeting of the formyl peptide receptors (FPRs) in myocardial infarction (MI). Specifically, the authors focus on identifying that FPR1/FPR2-biased agonism (i.e. the ability of GPCRs to adopt different conformational states, each linked to distinct cellular outcomes) may represent a novel therapeutic strategy for the treatment of this disease, and have suggested that small molecule FPR agonists with improved cardioprotective profile may be of benefit for treating myocardial infarction (MI).

B. Originality and interest: if not novel, please give references

This is an interesting paper, although lacks attention to detail in parts. Further experiments are needed in order to confirm the conclusions that have been drawn. If not, then it is not possible to draw the conclusions that authors state. Also, in a physiological setting, where only one dose is used, effects observed could simply be related to dose, rather than bias.

In response to Reviewer 2's concerns, we have now performed substantial additional experiments to both confirm the conclusions drawn (discussed in response to the itemised comments raised by Reviewer 2 at Section F below), and to further consider whether the lack of benefit observed with Cmpd43 administration simply relates to insufficient dose (as opposed to its lack of bias). The dose of Cmpd43 utilized *in vivo* was based on the dose previously used *in vivo* in a mouse model of acute ear inflammation (Ref#22 of the revised manuscript), as cited in the Methods. As detailed in Response to Reviewer 1 [Response 6 (ii)] and as illustrated in the revised manuscript (new Figure 5c), the plasma levels of Cmpd43 at 24 h and 48 h reperfusion *in vivo* suggest that the comparative lack of cardioprotection evident in response to Cmpd43 *in vivo* is not due to an insufficient dose, but rather is more readily explained by differences at the level of biased signalling away from Ca^{2+}_i . In line with these new findings, we have now revised our Discussion (please refer to the paragraph at the end of page 14, continuing into page 15).

Other new data referred to in the above summary are discussed in response to the itemised comments raised by Reviewer 2 at Section F below. We are confident that these new observations further strengthen our manuscript.

C. Data & methodology: validity of approach, quality of data, quality of presentation

Overall, as discussed below, several aspects of the data and methodology need to be fixed e.g. vehicle controls and inclusion of other FPR1 and FPR2 specific agonists needs to added to the study before the conclusion that FPR1/2 biased agonism represents a novel strategy to limit MI. Data presented also needs attention to detail, as discussed in the minor points below.

These aspects (e.g. vehicle controls, inclusion of additional FPR ligands, the appropriateness of our conclusions around FPR1/2 biased agonism, etc) are discussed in response to the itemised comments raised by Reviewer 2 at Section F below.

D. Appropriate use of statistics and treatment of uncertainties

Tests are fine.

E. Conclusions: robustness, validity, reliability

As discussed, without changes to the experimental design, it is not really possible to draw the conclusions that authors state.

As discussed both above at Section B, and below at Section F, we have now performed substantial additional experiments to specifically supplement our experimental design and hence strengthen the conclusion of this manuscript.

F. Suggested improvements: experiments, data for possible revision

These are itemised below under Major Concerns and Minor Comments in detail, as provided by Reviewer 2.

Major concerns:

- 1. Whilst the authors have stated that CHO cells do not express native Fprs (please provide reference), it would have been more relevant to use a human cell line e.g. HEK cells. It would be of benefit to repeat the key findings in HEK cells stably transfected with FPR1 or FPR2 as per the work of J. Gao and P. Murphy.**

CHO cells are ideally-suited for stable GPCR transfection, as only an extremely limited number of GPCRs are expressed in CHO cells, as we now confirm. We have performed gene expression analysis on CHO cells in this study, and could not detect native FPRs after 40 cycles of realtime PCR. The ABI Prism® 7700 Sequence Detection of quantitative real-time analysis consistently reported all 3 FPRs from this analysis as “undetected”. Furthermore in functional assays, FPR agonists do not elicit a response in CHO cells that have not been transfected with either hFPR1 or hFPR2. Collectively, these findings demonstrate that non-transfected CHO cells do not express native FPRs. Therefore, CHO cells stably transfected with human FPR1 or human FPR2 are ideally suited to observe and statistically evaluate FPR biased agonism at multiple signalling endpoints.

We are familiar with the FPR-transfected HEK studies published by Gao and Murphy, which provided excellent knowledge regarding FPR mechanisms particularly with respect to inflammatory cells (and cite one of these studies in the revised manuscript in response to comment 2 below). However, we do not believe that repeating our studies in hFPR1- and hFPR2-transfected HEK cells (an immortalised cell line originally derived from human embryonic kidney epithelia) would substantially strengthen the manuscript. Rather, we have elected to elucidate the impact of both Cmpd17b and Cmpd43 on Ca^{2+}_i in primary cardiomyocytes (a more relevant cell type for the study of MI), expressing native Fpr1 and Fpr2. These new data demonstrate that our conventional FPR-agonist Cmpd43 elicits concentration-dependent increases in Ca^{2+}_i ; in contrast, Cmpd17b elicits the opposite effect, yielding a concentration-dependent decrease in Ca^{2+}_i (new Figure 3c). This confirms that Cmpd17b elicits bias away from Ca^{2+}_i . in the most physiologically-relevant cell type, further strengthening the revised manuscript.

- 2. Whilst it is true that fMLP is a high affinity FPR1-selective peptide agonist, it does activate FPR2 at high concentrations. Therefore, it is critical to perform these**

experiments with an FPR1 specific agonist (note change needed on page 15 (see minor point 4 below). Also, add the data with the FPR2 specific agonist LXA4.

We agree with the reviewer that fMLP can activate FPR2 at high concentrations. However as evident in our data (Fig 2), fMLP has exquisite selectivity, with approximately 10 000-fold higher potency for FPR1 over FPR2. Importantly, the experiments involving fMLP were performed in hFPR1-, hFPR2-, or hFPR3-transfected CHO cells, each of which only expresses a single FPR subtype. Therefore, the ability of fMLP to activate FPR2 at high concentrations will not influence our results. Therefore, repeating our studies with an alternate FPR1 agonist would not influence our current interpretation of the results.

The data for LxA4 with respect to ERK1/2-phosphorylation are illustrated in Supplementary Figure 2: LxA4 failed to elicit any detectable response in either hFPR1-, hFPR2-, or hFPR3-transfected CHO cells; we have now observed an analogous lack of response on Ca^{2+}_i with LxA4 (see below). Similar findings have previously been observed by Hanson *et al* (Suppl Ref#1 of the revised manuscript supplement).

3. Is only FPR1 expressed on cardiomyocytes? What about Fpr2? It is unclear from the text.

We apologise for any confusion. Our evidence presented in Suppl Fig 2 now demonstrates that cardiomyocytes and cardiofibroblasts express all 3 major *Fpr* subtypes (noting that IUPHAR nomenclature for FPRs uses uppercase letters for human formyl peptide receptors, e.g. FPR1, whereas title case is used for rodent receptors, e.g. Fpr1; Ref#13). In cardiomyocytes, relative levels of expression of *Fpr1*>*Fpr2*>*Fpr3* are observed. Relative expression in cardiofibroblasts is *Fpr1*=*Fpr2*, with no detectable expression of *Fpr3*. In terms of functional response, we have previously demonstrated that *Fpr1* (but not *Fpr2*) is the subtype responsible for preserving myocardial viability and contractile function (Ref#8 of the revised manuscript). Both of our small-molecule agonists Cmpd17b and Cmpd43 are agonists to both receptor subtypes.

4. Supplementary figure 2, an n=1 is insufficient to draw conclusions. Further n numbers are needed.

We have now performed additional experiments to increase these time-course data to n=4 (hFPR1), with n=3 for hFPR2 and hFPR3, as illustrated in the revised Suppl Figs 2a-b, to specifically address this comment.

5. It was interesting to note that LXA4 failed to stimulate ERK-phosphorylation. - More discussion is needed here.

We have since tested an alternate batch of LxA4 obtained from a different source, and again failed to observe any ERK1/2 stimulation in hFPR2-transfected cells. Given the responses we observed to Cmpd43 in hFPR2-transfected CHO cells, we are confident that the transfected cells are fully functional and attribute the reproducible lack of response to LxA4 to the notoriously poor stability of LxA4 protein, and poor responsiveness at FPR2 (Ref#1 of the revised Supplement), as acknowledged above and on page 1 of the revised Supplement.

- 6. No vehicle was included in the figures, and needs to be include. This is important as especially as the vehicle for LXA4 is usually ethanol, and ethanol has been shown to induce e.g. ERK activation on its own (Christophe et al., 2002). Also, it would be important to confirm findings using a drug that does not signal through FPRs. Include the data.**

The Methods and Figure Legends for the *in vivo* components indicate that vehicle was administered to all “untreated” control mice subjected to LAD ligation. The vehicle controls for each agent (10% DMSO) were included in each study, but we neglected to clarify this in the original submission. We apologise for this oversight. The impact of vehicle control effects *in vitro* are now specifically illustrated in Supplementary Figure 2a-c.

- 7. Why were rat cardiomyocytes and mouse cardiac fibroblasts chosen, rather than both from the same species? As the *in vivo* methods were performed in mice it would be work validating the findings in mouse cardiomyocytes, rather than rat, or perform the *in vivo* work in rats, not mice.**

We acknowledge the species difference identified by Reviewer 2. Rat cardiomyocytes are the most commonly-utilised primary myocytes in research (due to their relative ease of isolation and reproducible yields and viability). While mouse cardiomyocytes are more technically-challenging, we have now performed complementary studies subjecting mouse cardiomyocytes to simulated I-R (New Fig 3b), ± Cmpd17b and Cmpd 43. These studies in mouse cardiomyocytes confirm the observations in rat cardiomyocytes, in which Cmpd17b (but not Cmpd43) significantly attenuates simulated I-R injury *in vitro*.

- 8. Results. Page 9 + Figure 4. Why was the dose 50mg/kg chosen for both compounds? Are the effects dose dependent? Include further doses. Also what happened with fMLP and LXA4 in these studies? Provide this data to strengthen the paper.**

The dose of 50mg/kg was chosen based on the dose of Cmpd43 used *in vivo* in a mouse model of acute ear inflammation (Ref#22 of the revised manuscript), as cited in the Methods.

Our new analyses confirm that this dose, administered every 24h (new Figure 5c), attains plasma levels of Cmpd43 at 24h and 48 reperfusion *in vivo* of ~1 μ M 20-24 h post final dose. The plasma levels of Cmpd43 *in vivo* are ~5-fold those of Cmpd17b at both timepoints, yet Cmpd43 is a more potent agonist than Cmpd17b at each of the 5 different signals determined in our *in vitro* studies (including all the kinase survival pathways, as well as Ca^{2+}_i). A higher dose of Cmpd43 would hence be unlikely to elicit cardioprotection. Further, our new analyses in primary cardiomyocytes confirm that bias at the level of Ca^{2+}_i is also evident in cells expressing native FPRs. Hence, as detailed in Response to both Reviewer 1 [Response 6 (ii)] and Reviewer 2 [Response to B above], the comparative lack of cardioprotection evident in response to Cmpd43 *in vivo* is more readily explained by differences at the level of biased signalling away from Ca^{2+}_i than the dose. Please refer to the paragraph at the end of page 14, continuing into page 15) for our revisions to the Discussion in line with these new findings.

We have not studied either fMLP or LxA4 *in vivo*, as Reviewer 2 suggested. Inclusion of either in our *in vivo* studies would be unlikely to have significant bearing on the manuscript findings, given the contrasting impacts already demonstrated in response to Cmpd17b versus Cmpd43. fMLP exhibits detrimental pro-inflammatory effects *in vivo* in a broad range pathologies (e.g. Taylor A, et al “SRF is required for neutrophil migration in response to inflammation” *Blood* 2016; 123:3027-36; Guo M, et al “Role of non-muscle myosin light chain kinase in neutrophil-mediated intestinal barrier dysfunction during thermal injury” *Shock* 2012; 38:436-43; Frommhold D et al, “RAGE and ICAM-1 differentially control leukocyte recruitment during acute inflammation in a stimulus-dependent manner” *BMC Immunology* 2011; 12:56), which would confound our measures in mice with LAD ligation and also would likely have detrimental impact on their wellbeing and raise animal ethics concerns. Further in our view, the lack of responsiveness to LxA4 *in vitro* precludes its utility in an *in vivo* context here.

9. Results. Page 9 + Supplementary Figure 4. These figures are poor. It is impossible to see whatever it is the arrows are pointing to. Neutrophils I assume?

The arrows in the representative immunofluorescent detection images in Supplementary Figure 4, were pointing to neutrophils (original Suppl Fig 4a, now Suppl Fig 5) and macrophages in the infarcted zone, respectively (original Suppl Fig 4b, now Suppl Fig 6). These images (which we uploaded in the specified resolution for the journal) represent lower magnification images at 20x, stitched together in a 9x9 format to illustrate the distribution of the relevant leukocyte populations in the myocardium after I-R. No quantitation has been performed on these images. Higher magnification representative images from the same samples, from which quantification was performed, are shown in revised manuscript Fig 5a (cardiac neutrophils) and new Suppl Fig 6 (cardiac macrophages).

10. What are the circulating counts for mice treated with these compounds? - Provide.

This data was reported in the original manuscript submission, Fig 6a-d. We have increased the n for this in the revised manuscript (see response to “11” below for specific changes to n values, and the impact of Cmpd17b and Cmpd43 on circulating cell counts).

11. n numbers throughout are extremely variable, especially in vivo (ranging from n=4-14). What is the reason for this?

We acknowledge that n for some *in vivo* data was low in the original submission. We have now performed additional experiments to specifically address this, with an additional n=3 sham (taking total shams to n=10 at 48h reperfusion) and an additional n=4 for all I-R groups (taking total vehicle I-R, I-R+Cmpd17b and I-R+Cmpd43 groups to n=9, 13 and 8 at 48h reperfusion, respectively) for circulating cell counts after LAD ligation *in vivo* (Fig 6a-d).

In the revised manuscript, each of total circulating WBC, circulating lymphocytes and circulating neutrophils are all significantly elevated by I-R, and all are significantly attenuated by Cmpd17b (but not by Cmpd43).

Although a modest tendency for I-R to increase circulating monocytes was evident, and a modest tendency for this to be blunted by Cmpd17b at 48h reperfusion, neither were statistically significant. In contrast, Cmpd43 not only failed to blunt circulating monocyte levels after I-R, but actually exaggerated this response, such that the combination of I-R and Cmpd43 significantly elevated the numbers of circulating monocytes compared to sham.

We have also increased CardioTACS measures of LV cell death at 7 days post I-R or sham to n=6-7/group.

12. Figure 3. Variability in results. Increase n numbers to 7 or 8, not 4.

We have now performed additional, complementary experiments in mouse cardiomyocytes (revised manuscript, Fig 3b), confirming our original observations in rat cardiomyocytes (Fig 3a), using the same species from which the *in vivo* data was obtained, but with less variability.

13. Do the compounds fMLP and LXA4 have any effect on differential circulating blood counts? Please provide all this data.

As discussed in detail in Response 8 above, we have not studied either fMLP or LxA4 *in vivo*.

14. Although plasma IL-1beta levels were measured, what about effects on other pro- and anti-inflammatory cytokines. Were any of these altered? Provide data.

We acknowledge that levels of other cytokines in response to LAD ligation ± Cmpd17b or Cmpd43 *in vivo* may be of interest. Only small plasma sample volumes can be collected from mice at study end. We elected however to prioritise determination of plasma levels of Cmpd17b and Cmpd43 (see responses to “B” and “8” above), in addition to measuring circulating WBC counts (see response to “11” above). However, given the requirement to determine plasma levels of Cmpd17b and Cmpd43, insufficient plasma sample remained for subsequent analyses such as impact on other circulating pro- and anti-inflammatory cytokines. We have however confirmed that Cmpd17b (but not Cmpd43) also blunted mouse cardiomyocyte *IL-1β* expression (Fig 3e, revised manuscript). Further, we now show in New Figure 8d-8g, that Cmpd17b blunts the MI-induced increase in LV gene expression of *Fpr1*, *Fpr2*, *CD68* and *TNF-α* in the infarcted zone of the LV (Figs 8d-8g, respectively).

15. n=5 for cells counts (I/R) was performed and yet n=8 for IL-1beta measurement. As the data is variable, it would be worth adding some more animals to the I/R group, especially as effects on circulating neutrophils and monocytes may be then seen.

As detailed in Response to “11” above, we have now performed additional experiments to specifically address this, increasing n by an additional 3-4 per group for circulating cell counts after LAD ligation *in vivo* (Fig 6a-d). The new data is shown in the updated Figure 6 and on page 11 of the revised manuscript, which reads:

“Circulating total WBC were significantly increased in vehicle-treated mice compared to sham (Fig 6a), with similar trends for each circulating lymphocytes and neutrophils. A similar but non-significant tendency for increases in both circulating pro-inflammatory IL-1β and circulating monocytes was also evident, consistent with systemic inflammation (Fig 6b-e). Cmpd17b significantly blunted I-R-induced increases in total WBC, circulating lymphocytes, neutrophils levels and IL-1β, without affecting circulating monocytes. In contrast, Cmpd43 significantly increased circulating monocytes and lymphocytes after 48h reperfusion compared to sham.”

16. Page 10 first paragraph states that "the I-R-induced increases in total WBC were and neutrophils were absent in Cmpd17b treated mice. Figure 6 shows a trend for these absent responses with Cmpd17b, but is not significant. Re-phrase this incorrect statement on page 10.

As detailed in Response to “11” and to “15” above, each of total circulating WBC, circulating lymphocytes and circulating neutrophils are all significantly elevated by I-R, and all are significantly attenuated by Cmpd17b (but not by Cmpd43).

17. Page 10 states that "taken together our observations suggest in contrast to Cmpd43, Cmpd17b attenuates the early inflammatory response associated with reperfusion injury". Could these effects simply be dose dependent?

As discussed in response to both Reviewer 1 (Response “6”) and Reviewer 2 (Response “B” above), our data suggests that the comparative lack of cardioprotection evident in response to Cmpd43 *in vivo* is not due to an insufficient dose, but rather is more readily explained by differences at the level of Ca^{2+}_i signalling downstream of FPRs. Indeed, in primary cardiomyocytes, Cmpd43 elicits concentration-dependent increases in Ca^{2+}_i ; in direct contrast to the opposite effect observed with Cmpd17b (new Figure 3c). Relative to Cmpd43, Cmpd 17b activation of cardioprotective signaling is not countered by concomitant potent increases in Ca^{2+}_i (which in the case of Cmpd43 likely negate its potential for cardioprotective actions on cardiac necrosis, inflammation and remodeling).

18. What happens at 24 h and 48 h on cardiac apoptosis. Do the compounds effect apoptosis at these time points?

At the suggestion of Reviewer 2, we have now performed additional experiments to specifically address impact on cardiac cell death at 48h reperfusion (to complement that already reported at 7-days reperfusion). This new data, shown in new Fig 5d, demonstrates that Further, the extent of LV cell death 48h post I-R was blunted by Cmpd17b (but not Cmpd43).

We have not undertaken comparable analyses after 24h *in vivo*, because this is likely to be too early a timepoint for detectable apoptosis to be fully developed in the myocardium.

19. There are FPR1 and FPR2 knock out animals. What happens with the compounds in these mice? It would strengthen the manuscript to include at least one.

Our data in CHO-cells stably-transfected with human FPR1 (but not FPR2), or CHO-cells stably-transfected with human FPR2 (but not FPR1) clearly implicate that both small-molecule FPR-agonists are acting as dual hFPR1/hFPR2 agonists (revised manuscript pages 5, 6, 14 and 16). Whilst Reviewer 2’s suggestion is of interest, we are not convinced that animals deficient in either *mfpr1* or *mfpr2* alone would significantly add to the manuscript. We await future development of mice deficient in both mouse *fpr1/fpr2* (if these are viable) but expressing human FPR1 for example, to compare the role of hFPR1 with hFPR2 in cardioprotection *in vivo*. Such an approach is beyond the scope of the present study.

20. Figure 6 shows circulating WBC and differential counts and plasma il-1b following 48 h reperfusion. What happens at 24h and 7 days? Include this data.

We have now increased the n for circulating cell counts 48h after LAD ligation *in vivo* (Fig 6a-d). The new data is shown in the updated Figure 6 and on page 11 of the revised manuscript.

The timepoint of 48h reperfusion was chosen for analysis of circulating WBC, differential cell counts and IL-1 β , as this equates to the approximate time of the peak in each of these responses post I-R (White DA, Su Y, Kanellakis P, Kiriazis H, Morand EF, Bucala R, Dart AM, Gao XM, Du XJ “Differential roles of cardiac and leukocyte derived macrophage migration

inhibitory factor in inflammatory responses and cardiac remodelling post myocardial infarction" *J Mol Cell Cardiol* 2014 69:32-42). The small volume of sample that can be collected from mice post I-R (~400µl) precluded measurement of cell counts and circulating IL-1β levels at 24h reperfusion, due to the volumes required for determination of plasma troponin analysis and plasma levels of FPR-agonists at 24h reperfusion. By 7 days reperfusion, levels of circulating WBC have declined by approximately 50% and circulating IL-1β levels have returned to levels observed in sham mice (White DA, *et al J Mol Cell Cardiol* 2014).

21. Figure 7. Images a+c. It would be helpful to include pictures at a higher magnification, as it is hard to see anything.

We apologise and have now included in the revised manuscript larger images for both Figure 7a and 7c in new Supplementary Figure 7a (which includes an inset highlighting the area at risk) and Supplementary Figure 7b.

22. Figure 7. Panel b: why n=4? It would be worth increasing these n numbers to equate with e.g. panel d. Also, sham in panel b looks as though there are only 3 values, not 4 as stated. Panel b.

We agree that n=4 for Figure 7a in the original submission was low, and have now performed additional analyses, increasing the numbers per group for CardioTACS at 7 days post I-R or sham to n=6-7, to specifically address this.

23. Figure 7. Is Cmpd17b not significant to sham, especially as I/R is?

For all results shown in Figure 7, Cmpd17b is not significantly different to sham (including cardiac cell death), as clarified in the text for each of cardiac cell death, collagen deposition, heart and LV weights (page 11).

24. Conclusions. Whilst this paper clearly focuses on myocardial models, very little discussion is provided on effects on other cells of these compounds, particularly in the in vivo setting. FPRs are expressed on a number of myeloid cells and endothelium. Discussion of the role that the different cell types play should be discussed and not excluded.

We agree that FPRs are indeed fairly ubiquitously expressed, and that bias away from Ca²⁺_i may also potentially be evident in myeloid cells for example, as it is in cardiomyocytes. We have now revised the Discussion in the revised manuscript to specifically address these comments, including an additional paragraph (Page 15) which reads: "*An ischemic insult stimulates expansion and mobilization of hematopoietic stem progenitor cells, to increase supply of myeloid cells to the injured myocardium*²⁹. The resultant increased cardiac accumulation of neutrophils and macrophages likely contribute to myocardial damage following the insult³⁰, as consistent with our observations 48h after I-R. Cmpd17b blunted several components of this inflammatory response, including total circulating levels of WBC, neutrophils and IL-1β, in addition to cardiac neutrophil density. FPRs, particularly FPR2, are expressed on myeloid cells and on endothelium; their activation also regulates the inflammatory response and its resolution³. We did not for example dissect out whether bias away from Ca²⁺_i, downstream of FPRs, was also evident in myeloid cells."

Minor comments:

- 1. Introduction (p5). More information regarding Cmpd43 and Cmpd17b should be provided e.g. Specific FPRs they bind to.**

We have included this in the introduction (see page 6). These agents were originally described as FPR1- (Cmpd17b) and FPR2-selective (Cmpd43), respectively. In the present study, both agents were however revealed as dual FPR1/FPR2 agonists, in contrast to their initial description as FPR1- (Cmpd17b) and FPR2-selective (Cmpd43), respectively.

- 2. FPR2 was previously known as 'FPR-like 1' not 'FPR-like' R. Amend accordingly**

We have amended this in the introduction (see page 5).

- 3. Introduction (p6) - remove the word 'hence' from paragraph two.**

We have amended this accordingly.

- 4. Page 15. fMLP is not FPR1 selective. It binds to FPR2 at lower affinity. This is misleading and should be amended.**

As detailed in response to Reviewer 2's Major Comment #2 above, our data clearly demonstrate that fMLP is robustly FPR1-selective, having 10 000-fold greater potency for FPR1 compared to FPR2 (Figs 1 and 2, Suppl Table 1). The term "FPR1-selective" used on page 15 remains the most appropriate terminology for this purpose.

- 5. Supplementary Figure 4. What are the arrows for? No mention in the figure legend.**

The arrows were intended to indicate Ly-6B.2-positive (neutrophils) or CD68-positive (macrophages) in the infarcted myocardium.

- 6. Figure 4a. These images are of poor quality. It would be worthwhile to include all slices taken for each heart.**

These images were uploaded in the specified resolution for the journal's Instructions to Authors. At the request of Reviewer 2, all slices for the representative hearts are now provided in revised Supplementary Figure 4.

- 7. Supplemental Figure 2D - Include line connecting dots as per other graphs.**

We have amended this accordingly.

- 8. Be consistent in the figures and manuscript. Either use capitals for each or lower-case i.e. A or a, but not a combination.**

In the revised manuscript, only lower case letters are used for Figure panels (i.e. a rather than A).

- 9. Figure 2 has two d's i.e. cAMP should be image e, not d.**

We apologise for this mistake, which we have corrected in the revised manuscript; Figure 2e refers to cAMP.

10. Page 8, first paragraph, remove comma after GPCR.

We have amended this accordingly (now top of page 9).

11. All figure legends should have doses/concentrations of the drugs used.

The concentrations for all drugs used in Figures 1 and 2, as well as Supplementary Figure 2d-2f are shown on the X axis for each panel. We have amended Figure Legends for Figures 3-8, as well as Supplementary Figures 2a-2c, 3b-f, 4, 5, 6 and 7 where required, to include doses/concentrations according to reviewer 2's request.

12. Page 10 second paragraph, Cmpd17b is missing the 'b'.

We have amended this accordingly (now top of page 11).

13. Supplemental. Where are Cmpd17b and Cmpd43 obtained from?

Cmpd17b and Cmpd43 were synthesized by Anthem Bioscience, Bangalore, India, as detailed in the Methods (page 18). We have moved this information regarding synthesis to the first paragraph of the Methods (instead of the Supplement) for reader clarity.

14. Figure 5 a+c. These figures are unclear, either clearly emphasise the point (e.g. add magnified images) or remove.

We have improved the clarity of both Figure 5a and the original Figure 5c (which has now been moved from the main figures to Supplementary Fig 6) in the revised manuscript.

G. References: appropriate credit to previous work?

References are fine.

H. Clarity and context: lucidity of abstract/summary, appropriateness of abstract, introduction and conclusions

These are fine.

REVIEWER 3

Remarks to the Author.

A. Summary of key results:

In this study, Qin et al. report that the FPR1/2 agonist Cmpd17b is biased against the induction of Ca²⁺ signaling, while retaining P-ERK and P-Akt activation, versus the conventional agonist Cmpd43, suggesting that it could promote survival signaling in the absence of Ca²⁺-mediated cell death and inflammation. Indeed, 17b reduces H-R-induced cardiomyocyte toxicity as well as TGFβ-induced inflammatory gene expression in cardiac

fibroblasts in vitro. Further, 17b was shown to reduce the injury and immune responses to I-R in vivo leading to altered cardiac remodeling.

No specific issue to be addressed raised here.

B. Originality and interest:

Overall the study offers exciting new insight into the potential therapeutic utility of biased ligands, particularly for FPR, in achieving better outcomes following cardiac injury.

No specific issue to be addressed raised here.

C. Data and methodology:

Overall the methodology is performed, and data presented, well. However:

1. Peak cell death after ischemic injury occurs rapidly and subsides over time, thus it's unclear why the authors measured TUNEL at 7 days post-I-R versus using their 24-hr timepoints when the bulk of the cell death will have occurred and larger differences in 17b-mediated protection would be observed.

We agree that the peak timepoint of necrotic cell death following a myocardial ischemic insult is 24h. Importantly, and consistent with Reviewer 3's comments, our data clearly demonstrate that Cmpd17b mediates significant cardioprotection at this timepoint in the intact heart *in vivo*, as demonstrated at the level of infarct size and plasma cTnl (Figs 4a, 4c, 4d), conventional markers of cardiac necrosis. However, multiple forms of cardiomyocyte death contribute to final infarct size, with necrosis and apoptosis (and potentially also autophagy, necroptosis and pyroptosis) can continue in the previously ischemic area for several days post-reperfusion (Bell RM *et al* "Position document on ischaemia/reperfusion injury, conditioning and the ten commandments of cardioprotection" *Basic Res Cardiol* 111: 41, p2-13, 2016; Baines CP "How and when do myocytes die during ischemia and reperfusion: the late phase" *J Cardiovasc Pharmacol Ther* 16: 239-43, 2011). Indeed, apoptosis is apparent at day-7 post IR, as shown in the present study.

Cmpd17b mediates significant cardioprotection at 7-days post I-R in the intact heart *in vivo* using our CardioTACS detection of LV cell death, as can be observed in Figs 7a, 7b and Suppl Fig 7a. In the revised manuscript, we have increased n for the 7-day timepoint. Further, in response to Reviewer 3's comments, we have also performed analyses of CardioTACS detection of LV cell death at 48h reperfusion (new Fig 5d). We have not undertaken CardioTACS detection of LV cell death 24h after I-R *in vivo*, firstly because, in mice allocated to the 24h reperfusion time-point, the entire myocardium is subjected to Evans blue dye in the intact mouse heart *in situ* followed by excision of the heart for TTC staining, for determination of myocardial necrosis. It is thus not possible to examine these tissues at this timepoint for other analyses. Secondly, the TTC analysis and the cTnl determined at this timepoint already provided two measures of cell death after 24h reperfusion.

Hence, we now provide measures of cell death (and the impact of Cmpd17b) at 3 timepoints post I-R, 24h (necrosis), 48h and 7-days (both DNA fragmentation, i.e. apoptosis, via CardioTACS).

D. Statistics:

1. Authors should define the statistical significance and test used for Figs 1F and 2F in the legends.

We have included this in the Figure Legends for Fig1f and 2f as requested. Specifically we state “* $P < 0.05$ vs Cmpd43 on the $\Delta \text{Log}(\tau/\text{KA})$ from Supplemental Table 1, via one-way-ANOVA with Dunnet’s post-hoc test.”

2. In the results section, referring to the data in Fig. 7D the authors state that "the impact of I-R on LV fibrosis in this region was absent in Cmpd17b-treated mice", which based on the statistics is not true.

We have clarified the wording of this section, on page 12 of the revised manuscript, which now reads “Although the impact of I-R on LV fibrosis in this region persisted in Cmpd43-treated I-R mice, cardiac collagen deposition in Cmpd17b-treated mice was not significantly different from either sham or vehicle I-R (Fig 7d)”.

E. Conclusions:

1. A significant study limitation is that the signal bias of 17b shown in exogenous cells has not been linked to its impact on in vitro and in vivo injury models. 17b is clearly more protective than cmpd43, but the proximal signaling responses in these models have not been investigated.

To specifically address this comment, we have now performed concentration-response studies to both FPR-agonists with respect to cardiomyocyte intracellular Ca^{2+} mobilization (Ca^{2+}_i), in primary cardiomyocytes. These data demonstrate that our conventional FPR-agonist Cmpd43 elicits concentration-dependent increases in Ca^{2+}_i ; in contrast, Cmpd17b elicits the opposite effect, yielding a concentration-dependent decrease in Ca^{2+}_i (new Figure 3c). This confirms that Cmpd17b elicits bias away from Ca^{2+}_i , in primary cardiomyocytes (expressing native FPRs), as well as in CHO cells stably expressing the human FPR subtypes. We believe these new data counter the perceived limitation and further strengthen the manuscript and its conclusions.

2. Some of the statements regarding the ability of 17b to block effects versus I-R condition in vivo are too strong considering the data does not appear statistically significant (fibrosis, immune cell levels).

We acknowledge that some observations were underpowered in the original submission, and have thus now performed additional experiments to improve this. Specifically:

- Increasing circulating cell counts after 48h reperfusion by additional $n=3$ shams (taking total shams to $n=10$ at 48h reperfusion) and additional $n=4$ I-Rs for all I-R groups (taking total vehicle I-R, I-R+Cmpd17b and I-R+Cmpd43 groups to $n=9$, 13 and 8 at 48h reperfusion, respectively) after LAD ligation *in vivo* (Fig 6a-d)
- Increasing CardioTACS measures of LV cell death at 7 days post I-R or sham to $n=6-7$ /group (Fig 7b).
- Including CardioTACS measures of LV cell death at 48h post I-R or sham (new Fig 5d).

With increased group size in the revised manuscript, each of total circulating WBC, circulating lymphocytes and circulating neutrophils are significantly elevated by I-R, and all are found to be significantly attenuated by Cmpd17b (but not by Cmpd43). Cmpd17b did not affect circulating

monocytes. In contrast, Cmpd43 significantly increased circulating monocytes and lymphocytes after 48h reperfusion compared to sham.

Further, regarding the impact of Cmpd17b on LV fibrosis after 7-days I-R: we have changed the wording of these results, to now read “Although the impact of I-R on LV fibrosis in this region persisted in Cmpd43-treated I-R mice, cardiac collagen deposition in Cmpd17b-treated mice was not significantly different from either sham or vehicle I-R (Fig 7d).” (see page 12).

F. Suggested Improvements:

1. Does the signal bias translate to primary myocytes and fibroblasts? While in theory it should, it is important to show that at least the difference in Ca²⁺ responsiveness with preserved P-ERK/Akt signaling is recapitulated in these cells, thereby linking the signal bias data in exogenous systems to the measured outcomes post-H-R and post-TGFβ in the endogenous primary cell models of interest. The authors have previously reported that FPR1 in particular is responsible for the cardioprotective benefit of the FPR ligands ANX-A1 and Ac-ANX-A1(2-26) (Qin et al. 2013), however it would be of benefit to show in the isolated myocyte and fibroblast models that the effects of 17b via endogenous FPRs can indeed be blocked with an FPR antagonist.

We agree with this view and have now performed new experiments to specifically address these comments. This includes concentration-response studies to both FPR-agonists with respect to cardiomyocyte intracellular Ca²⁺ mobilization (Ca²⁺_i), in neonatal rat cardiomyocytes, which demonstrate that our conventional FPR-agonist Cmpd43 elicits concentration-dependent increases in Ca²⁺_i; in contrast, Cmpd17b elicits the opposite effect, yielding a concentration-dependent decrease in Ca²⁺_i (new Figure 3c). This confirms that Cmpd17b elicits bias away from Ca²⁺_i. We believe these new data further strengthen the manuscript and its conclusions.

2. One of the pieces of data for the I-R studies that is not shown is the plasma level of each of the compounds measured. The authors state in the discussion that they were measured 7 days after treatment in vivo and were not different, but in the supplemental methods state they were measured several hours after dosing. Are the compounds really still present after 7 days, and if so is this a relevant timepoint for comparison of bioavailabilities? Given the timeframe of acute cell death and inflammation that occurs immediately after I-R, it would make more sense to report the plasma levels of each compound around the time of surgery when enhanced survival signaling would be expected to dampen the initial cell death response. This information is critical and the values should be shown since differences in the injury and immune responses could result from different pharmacokinetics if the timing of intervention is essential. It's also critical to show empirically that Cpd43 was found at similar levels to 17b since, as the authors note, Cpd43 did not produce any significantly different effects from I-R alone.

To clarify, both Cmpd17b and Cmpd 43 were administered at 50mg/kg daily i.p. commencing at the end of the surgical LAD occlusion, just prior to reperfusion. Plasma concentrations were determined at endpoint, ~20h after the final dose. The limit of detection of both FPR-agonists in plasma of 1ng/ml (now stated on p27 of the revised Methods); hence both compounds are still present at endpoint. At the suggestion of Reviewer 3, we now report plasma levels of both compounds at 24h and 48h after reperfusion (new Fig 5c), the timepoints at which cardiac injury (necrosis and inflammation in particular) were likely maximal. These data suggest that the comparative lack of cardioprotection evident in response to Cmpd43 *in vivo* is not due to an insufficient dose, but rather is more readily explained by differences at the level of biased

signalling away from Ca^{2+}_i . (consistent with our new results demonstrating that the conventional FPR-agonist Cmpd43 elicits concentration-dependent increases in Ca^{2+}_i in primary cardiomyocytes, in direct contrast to the opposite effect in response to Cmpd17b), as detailed above, and illustrated in new Figure 3c. We have not sought more detailed pharmacokinetic information earlier than the primary experimental endpoint, as animal ethics constraints preclude blood collection from living mice in the first 48h following open chest surgery (due to risk of sudden death as a result of handling), unless the collection is a non-recovery procedure from anesthetized animals.

3. Cardiac function is not shown, which is a major limitation: although 17b decreases cell death and inflammation, the ultimate therapeutic goal is to preserve or improve cardiac contractility over time (several weeks post-I-R). This is also important to know based on the gravimetric data wherein the 17b-treated I-R hearts appear to become smaller than sham hearts after 7 days. What is causing this effect, and are the hearts functioning normally? Since there are no sham + cmpd controls, the effect of 17b in the context of I-R cannot be separated from any long-term impact on cardiac structure it may have in the absence of injury.

We have now performed considerable experiments to specifically address these comments. Firstly, as requested by Reviewer 3, we have now investigated the potential for Cmpd17b cardioprotection at the level of contractile function over the short- and longer-term. In this follow-up study, mice in Cohort 4 were subjected to 4-weeks MI. Cardiac function was determined via M-Mode echocardiography at baseline (prior to surgery) and at both 7- and 28-days post MI. As shown in new Fig 8, Cmpd17b significantly preserved both fractional shortening and velocity of circumferential fiber shortening (Vcfc), demonstrating that Cmpd17b cardioprotection over the short-term indeed translated into functional benefit several weeks after the ischemic insult (see page 12).

Reviewer 3 also commented on the endpoint heart:body weight ratio (HW:BW) in mice subjected to I-R \pm Cmpd17b. As shown in Supplementary Table 2, HW:BW after 7-days I-R was 4.6 ± 0.2 , 5.2 ± 0.2 and $4.0 \pm 0.1 \text{ mg/g}$ in sham, vehicle-treated and Cmpd17b-treated I-R mice respectively. Cmpd17b-treated I-R mice significantly attenuated the I-R-induced increase in HW:BW at this timepoint. This apparent tendency for a smaller heart following Cmpd17b administration after 7-days I-R was not significantly different to heart size in shams at the same timepoint, and was not evident after 48h I-R (Supplementary Table 2), nor after 4wks MI (see new Supplementary Table 3 in the revised manuscript). We are hence confident that 17b-treated I-R hearts are not detrimentally impacted by Cmpd17b in terms of both cardiac size and cardiac function over the longer-term.

G. References: No concerns

H. Clarity and Context:

- 1. The authors do not state whether they think that the effects of 17b on immune cell infiltration into the heart (and circulating levels of immune cells) are due to the decreased injury itself, or direct effects on immune cells themselves. This interpretation could change the importance of the data. For instance the decreased infiltration and immune cell levels in blood are consistent with reduced injury and could be presented more clearly as such. However, if the authors think 17b impacts the immune cells themselves, then additional in vitro**

migration/chemotaxis assays could be performed to strengthen this idea since none of the blood cell data is statistically significant as reported (17b vs I-R).

We completely agree with this important point. As detailed in response to E2 above, the additional experiments we have undertaken for the manuscript revision clearly demonstrate that each of total circulating WBC, circulating lymphocytes and circulating neutrophils are all significantly elevated by I-R, and all are significantly attenuated by Cmpd17b (but not by Cmpd43). Cmpd17b also significantly blunts circulating IL-1 β . As Reviewer 3 suggests, this is indeed consistent with a reduced injury. We did not dissect out whether Cmpd17b's ability to directly limit the infarct size *in vivo*, and hence the inflammatory stimulus, was the sole mechanism of cardioprotection *in vivo*, or whether bias away from Ca²⁺_i was also evident in myeloid cells (providing a potential additional mechanism of protection). Importantly we have now confirmed that Cmpd17b (but not Cmpd43) favors downstream signalling away Ca²⁺_i specifically in cardiomyocytes. We have thus included an additional paragraph in the revised manuscript Discussion considering the potential for direct impact on immune cells (see Page 15, detailed below), to specifically address this comment, as well as revising other phrases in the Discussion (as highlighted by red underline in the marked up manuscript).

The additional paragraph reads: "*An ischemic insult stimulates expansion and mobilization of hematopoietic stem progenitor cells, to increase supply of myeloid cells to the injured myocardium²⁸. The resultant increased cardiac accumulation of neutrophils and macrophages likely contribute to myocardial damage following the insult²⁹, as consistent with our observations 48h after I-R. Cmpd17b blunted several components of this inflammatory response, including total circulating levels of WBC, neutrophils and IL-1 β , in addition to cardiac neutrophil density. FPRs, particularly FPR2, are expressed on myeloid cells and on endothelium; their activation also regulates the inflammatory response and its resolution³. We did not for example dissect out whether bias away from Ca²⁺_i, downstream of FPRs, was also evident in myeloid cells.*"

REVIEWERS' COMMENTS:

Reviewer #1 (Remarks to the Author):

The Authors have addressed my concerns and replied to my suggestions. As such I am quite pleased that the revised manuscript is markedly improved and delivers a clear scientific message. Experiments are plenty and the conclusions are solid. I have minor suggestions the Authors may want to take on board:

1. Page 8 Lines 135-136 and Page 9 Line 137. I would remove the data generated with LXA4. They do not add to the manuscript and study and they are presented only in supplementary figure. They are negative too. The reason for my request is that there is a bit of confusion on LXA4 handling and biology with highly negative papers where the bioactive lipid mediator has not been properly handled or indeed solubilised (e.g. DMSO destroys the lipid). As such, it is now customary to present the Mass Spect of the LXA4 used to demonstrate the quality of the product used. I think this is too much to ask herein, also in view of the extremely peripheral importance of the experiments with LXA4. I think it is much better to remove them, and remove the relevant references from the Reference List.
2. Page 11, Line 205, I would remove 'each'.
3. Page 12, Lines 219-222. Rephrase this sentence because a superficial read suggests that Cpd43 is active, whereas the message is the opposite.
4. Page 12, Line 226: I am not sure it is wise to refer to a cohort 4 here.
5. Page 14, Line 273: I would write '...(administered just prior to reperussion)...'
6. Page 15, Line 291. Herein the Authors may want to insert a hypothesis on why the Ca²⁺ increase evoked by Cpd43 negates its potential to confer cardioprotection. In my view the study is complex and full of experimental approaches so I am not asking for more, however hypotheses can be out forward: e.g. text on Page 16 Line 318-319 could be moved.

Reviewer #2 (Remarks to the Author):

Qin et al have revised their manuscript well, and their findings will add value to the FPR field. No further issues remain, other than some grammatical ones, which will be picked up during editing for submission to the journal.

Reviewer #3 (Remarks to the Author):

All points raised in the previous round of review have been satisfactorily addressed. The authors are commended on their thorough study into the biased nature of the FPR agonist compound 17b and its ability to protect the heart against ischemia-reperfusion injury-induced damage. Statistics are appropriate and methodology comprehensive.

“Small Molecule-Biased Formyl Peptide Receptor Agonist Compound 17b Protects Against Myocardial Ischemia-Reperfusion Injury in Mice”

Cheng Xue Qin, Lauren May, Renming Li, Nga Cao, Sarah Rosli, Minh Deo, Amy Alexander, Duncan Horlock, Jane Bourke, Yuan Yang, Alastair Stewart, David Kaye, Xiao-Jun Du, Patrick Sexton, Arthur Christopoulos, Xiao-Ming Gao, and Rebecca Ritchie.

RESPONSE TO REVIEWERS AND EDITOR

We are delighted by the positive response from all three Reviewers, and have responded the remaining suggestions into the revised manuscript. The Reviewers' comments are provided in italics in response to reviewers, with our response immediately following.

We have endeavoured to ensure our revised manuscript complies with the editorial policies of Nature Communications, and have completed and uploaded the checklist for formatting requirements with the revised article, in addition to the completed checklist for reporting requirements. In particular, the data availability statement on page 26 of the revised manuscript states that the data that supports the finding of this study are available from the corresponding author on request.

Reviewer #1 (Remarks to the Author):

The Authors have addressed my concerns and replied to my suggestions. As such I am quite pleased that the revised manuscript is markedly improved and delivers a clear scientific message. Experiments are plenty and the conclusions are solid. I have minor suggestions the Authors may want to take on board:

1. Page 8 Lines 135-136 and Page 9 Line 137. I would remove the data generated with LXA4. They do not add to the manuscript and study and they are presented only in supplementary figure. They are negative too. The reason for my request is that there is a bit of confusion on LXA4 handling and biology with highly negative papers where the bioactive lipid mediator has not been properly handled or indeed solubilised (e.g. DMSO destroys the lipid). As such, it is now customary to present the Mass Spect of the LXA4 used to demonstrate the quality of the product used. I think this is too much to ask herein, also in view of the extremely peripheral importance of the experiments with LXA4. I think it is much better to remove them, and remove the relevant references from the Reference List.

We thank Reviewer 1 for their comments. We have revised the text as suggested, and remove the main text, reference and supplementary data (figure 2 a, b, c, e) containing LxA4.

2. Page 11, Line 205, I would remove 'each'.

We have removed each as reviewer suggested.

3. Page 12, Lines 219-222. Rephrase this sentence because a superficial read suggests that Cpd43 is active, whereas the message is the opposite.

We have rephrased the sentence as following:

The impact of I-R on LV fibrosis in this region persisted in Cmpd43-treated I-R mice, **however** cardiac collagen deposition in Cmpd17b-treated mice was not significantly different from either sham or vehicle I-R (Fig 7d).

4. Page 12, Line 226: *I am not sure it is wise to refer to a cohort 4 here.*
The impact of Cmpd17b on ischaemia-induced cardiac dysfunction and inflammatory gene expression were assessed 4 weeks after permanent LAD occlusion (i.e. in the setting of MI rather than I/R).

5. Page 14, Line 273: *I would write '...(administered just prior to reperfusion)...'*

We have revised as reviewer suggested, which now reads:

Cmpd17b (administered just prior to reperfusion) significantly reduced early cardiac necrosis, consistent with promotion of cardiomyocyte survival, in contrast to Cmpd43 at the same dose.

6. Page 15, Line 291. *Herein the Authors may want to insert a hypothesis on why the Ca²⁺ increase evoked by Cpd43 negates its potential to confer cardioprotection. In my view the study is complex and full of experimental approaches so I am not asking for more, however hypotheses can be out forward: e.g. text on Page 16 Line 318-319 could be moved.*

We have revised each as reviewer suggested as below, moving the hypothesis to this location in the text as suggested by Reviewer 1. The sentence reads:

Relative to Cmpd43, Cmpd17b activation of cardioprotective signalling is not countered by concomitant potent increases in Ca²⁺_i (which in the case of Cmpd43 likely negate its potential confer cardioprotection). We hypothesised that if FPR agonists with biased signalling profiles towards (rather than away from) Ca²⁺_i are generated endogenously during MI, then Cmpd17b may act to block Ca²⁺-overload by preventing receptor activation by such putative endogenous ligands. This is however beyond the scope of this study.

Reviewer #2 (Remarks to the Author):

Qin et al have revised their manuscript well, and their findings will add value to the FPR field. No further issues remain, other than some grammatical ones, which will be picked up during editing for submission to the journal.

Response: We thank Reviewer 2 for their comments. The Reviewer raises no further issues to address regarding this work.

Reviewer #3 (Remarks to the Author):

All points raised in the previous round of review have been satisfactorily addressed. The authors are commended on their thorough study into the biased nature of the FPR agonist compound 17b and its ability to protect the heart against ischemia-reperfusion injury-induced damage. Statistics are appropriate and methodology comprehensive.

Response: We thank Reviewer 3 for their comments. The Reviewer raises no further issues to address regarding this work.

Editor (Remarks to the Author):

1) *Please ensure that your manuscript follows the formatting guidelines described in:*

<http://www.nature.com/ncomms/submit/content-types>

We have uploaded the revised manuscript as a Word file, indicating where all changes have been made by using the Tracked Changes function and leaving the Editor's comments in. We have endeavoured to ensure our revised manuscript complies with the editorial policies of *Nature Communications*. This has included methods in the main text. Our supplement now only includes Table, Figures and the accompanying supplementary data description. The final revised manuscript remain under the word limit.

All figures have been uploaded as .psd format as requested. Their apparent resolution in the Pdf during manuscript submission is not as high resolution as the .psd version already uploaded. We are happy to email these directly to you if required.

2) *I have made a minor edit of your abstract so that it complies with our journal's style. We accept the editors' minor changes of the abstract.*

3) *We do not have "abbreviations" section, so all abbreviations should be defined upon first mention in the text.*

We removed the abbreviation section as suggested by the editor, and define all abbreviations at first mention in the text.

4) *Please make sure that all figures, main and supplementary, are referred to at least once in the manuscript. Also, please make sure that the first referral to any given figure is in numerical order (i.e. you should not be referring to Figure 2 before you mention Figure 1.*

We have checked all the figures in the main text and supplementary are referred to in the manuscript, and appear in the text in a logical order.

5) *When stating the p value in the text, please define the statistical test used to calculate this p value. Please correct wherever it applies.*

We have added the statistical analysis after the p value in the Results section, as per page 6-10 of the final manuscript.

6) *Latin words should be in italic.*

We have changed the Latin words to italic through the text (e.g. VS on P8-12).

7) *We don't allow for subheading nesting- only one tier of subheadings is allowed. Please correct wherever it applies.*

We have removed the numbers in front of the subheadings in the Methods section, thus only one tier of subheadings are presented.

8) *Please specify all primer sequences used in this study. If there are many primers, you may want to consider generating a Supplementary table with all primer sequences.*

We have included the primer sequence in supplementary Table 4 of the revised manuscript.

	Forward primer	Reverse primer
18S	TGTTCCACCATGAGGCTGAGATC	TGG TTG CCT GGG AAA ATC C
mβ-MHC	TCT CCT GCT GTT TCC TTA CTT GCT A	GTA CTC CTC TGC TGA GGC TTC CT
mCTGF	TGA CCC CTG CGA CCC ACA	TAC ACC GAC CCA CCG AAG ACA CAG
mTGFβ	TGGAGCAACATGTGGAAGCTC	GTCAGCAGCCGGTTACCA
mIL-1β	TTGACGGACCCCAAAGAT	GAAGCTGGATGCTCTCATCCTG
mFPR1	CCTTGGCTTTCTTCAACAGC	GCCCGTTCTTTACATTGC AT
mFPR2	ACA GCA GTT GTG GCT TCCTT	CCT GGC CCA TGA AAA CAT AG
mCD68	CCAATTCAGGGTGGAAGAAA	CTCGGGCTCTGATGTAGGTC
mTNFα	CTG TAG CCC ACG TCG TAG C	TTG AGA TCC ATG CCG TTG

Supplementary Table 4.

5'-3' primer sequences used for gene expression by qrt-PCR.

9) *Provenience of all cell lines used in this study should be specified.*

The provenience of all cell lines used in the study has been specified in the final manuscript.

CHO cell cultures were tested routinely for mycoplasma contamination. Human FPR (*hFPR1*, *hFPR2* and *hFPR3*) sequences were amplified by PCR and cloned into the Gateway entry vector, pDONR202TM, using the BP clonase enzyme mix, according to the manufacturers' instructions (refer to supplementary table 5 for primer sequence). The *hFPR* pDONR201TM constructs were subsequently transfected into the pEF5/FRT/V5/dest vector using the LR clonase enzyme mix (Invitrogen). Receptor constructs (pEF5/FRT/V5/dest) were used to transfect Flp-In CHO cells (Invitrogen)³³.

Supplementary Table 5

	Gateway entry vector 5'	Stop in gateway entry vector 3'
FPR1	ggggacaagttgtacaaaaaagcaggctt cCACCATGGAGACAAATTCT CTCTCCCACG	ggggaccactttgtacaagaaagctgggtcTCACTTTG CCTGTAACCTCCACCTCTGC
FPR2	ggggacaagttgtacaaaaaagcaggctt cCACCATGGAAACCAACTTCT CCTACTCC	ggggaccactttgtacaagaaagctgggtcTCACATTG CCTGTAACCTCAGTCTCTGC
FPR3	ggggacaagttgtacaaaaaagcaggctt cCACCATGGAAACCAACTTCT CCATTC	ggggaccactttgtacaagaaagctgggtcTCACATTG CTTGTAACCTCCGTCTCCTC

Gateway entry vector used for generating the hFPR cell line. The capital letters reflect the FPR sequence whereas the small letters are the gateway entry vector sequence required to insert the gene into the gateway entry vector plasmid.

10) Please provide sufficient detail. We strive to achieve complete protocol transparency so that each experiment can be reproduced by others and thus do not allow for the use of "as described previously." This applies to all instances where "as described" or similar is used.

We have included more information on NRCM isolation in particular, as per below

Isolation of primary neonatal rat cardiomyocytes (NRCM) and adult mouse cardiac fibroblasts (AMCF) was performed by serial enzymatic digestion. Neonatal rats were decapitated and hearts were immediately placed in Hanks Balanced Salt Solution (HBSS, 9.5g per L Hanks Balanced Salts, 350 mg per L sodium bicarbonate). Ventricles were removed and digested in trypsin (1mg per ml trypsin in HBSS) overnight at 4°C and collagenase (2.5 mg per heart) four times at 37°C for 10 min. Cardiomyocytes were suspended in sterile DMEM, supplemented with penicillin 100U/mL, streptomycin 100µg/mL and 10% FBS. The cells were pre-plated twice (45min at 37°C) to reduce fibroblast contamination. AMCF were obtained as a by-product of cardiomyocyte isolation.

11) All mouse genes should be written in italic with only the first letter in upper case. Read more below.

We have revised as editor suggested. We included the statement on page 26 of the final manuscript.

"The data that supports the findings of this study are available from the corresponding author on request."

12) Please make Data Availability a separate subheading.

We have revised this as per the editor's suggestion.

13) Source and funding should be part of Acknowledgments.

We have revised as editor suggested.

14) Each figure legend should not be longer than 350 words, its title should not be longer than one line in a word doc. and it should contain:

*n numbers and number of replicates

*p values and name of statistical test

*definition of error bars

*scale bar length

*definitions of symbols/colors and new abbreviations

We have included all necessary information as the editor suggested.

15) Scale bar missing for Figure 4. Please add and define its length in the figure legend

We have revised the Figure as editor's suggestion.

16) Please check whether your manuscript contains third-party images, such as figures from the literature, stock photos, clip art or commercial satellite and map data. We strongly discourage the use or adaptation of previously published images, but if this is unavoidable,

please request the necessary rights documentation to re-use such material from the relevant copyright holders and return this to us when you submit your revised manuscript.

Our manuscript does not have third-party images.

17) Your paper will be accompanied by a two-sentence editor's summary, of between 250-300 characters, when it is published on our homepage. Could you please approve the draft summary below or provide us with a suitably edited version.

We are happy with the two sentence summary.

G Protein-Coupled Receptors (GPCRs) can adopt different conformations, each linked to distinct cellular outcomes. Here the authors show that compound 17b, a novel agonist of the GPCR family member FPR, robustly activates cardioprotective but not detrimental FPR signalling, showing beneficial therapeutic effect in a mouse model of cardiac infarction.